# Implementation of a Simple Actuator Disc for Large Eddy Simulation in the Weather Research and Forecasting Model (WRF-SADLES-V1.2) for Wind Turbine Wake Simulation

Hai Bui[1,2], Mostafa Bakhoday-Paskyabi[1,2], and Mohammadreza Mohammadpour-Penchah[1,2]

[1]Geophysical Institute, University of Bergen, Allégaten 70, 5007 Bergen, Norway
[2]Bergen Offshore Wind Centre, Allégaten 55, 5007 Bergen, Norway

**Correspondence:** Hai Bui (hai.bui@uib.no)

**Abstract.** In this study, we present the development of a Simple Actuator Disc model for Large Eddy Simulation (SADLES), implemented within the Weather Research and Forecast (WRF) model, which is widely used in atmospheric research. The WRF-SADLES model facilitates both idealized and realistic downscaling of large eddy simulations, with a focus on resolutions of tens of meters. Through a comparative analysis with the Parallelized Large-eddy Simulation Model (PALM) at resolutions of 10 meters and 30 meters, we validate the effectiveness of WRF-SADLES in simulating the wake characteristics of a 5-MW wind turbine. Results indicate good agreement between WRF-SADLES at 30-meter resolution, 10-meter resolution, and the PALM model. Additionally, we demonstrate a practical case study of WRF-SADLES by downscaling ERA5 reanalysis data using a nesting method to simulate turbine wakes at the Alpha Ventus wind farm in the South of the North Sea. The meso-to-micro downscaling simulation reveals that the wake effect simulated by WRF-SADLES at the FINO1 offshore meteorological mast station aligns well with the cup anemometer and LiDAR measurements. Furthermore, we investigate an event of farm-to-farm interaction, observing a 16% reduction in ambient wind speed and a 38% decrease in average turbine power at Alpha Ventus due to the presence of a wind farm to the southwest. WRF-SADLES offers a promising balance between computational efficiency and accuracy for wind turbine wake simulations, making it valuable for wind energy assessments and wind farm planning.

## 1 Introduction

Wind energy has become increasingly important due to the pressing need to address climate change and reduce our reliance on fossil fuels. It is crucial to have a deep understanding of the complex interaction between wind turbines and the atmospheric boundary layer (ABL) for effective wind energy assessment and wind farm construction planning. In particular, the wake created behind upstream turbines can have a significant impact on the performance of downstream turbines, resulting in reduced power production, increased turbulence, and dynamic loading (Porté-Agel et al., 2020; Bakhoday-Paskyabi et al., 2022a). Large eddy simulation (LES) is a powerful tool that can simulate ABL turbulence with high spatial and temporal resolution, making it an invaluable resource for studying turbine wakes and and their interaction with the ABL (Breton et al., 2017).

To model the turbine wakes, various methods can be used to simulate the rotor with varying levels of accuracy and computational cost. The most accurate, but computationally expensive methods are the actuator surface (AS) and actuator line (AL) models, which typically use the Blade Element Momentum (BEM) theory or the Blade Element Theory (BET) (Burton et al., 2011; Göçmen et al., 2016) to calculate the thrust and torque acting on the turbine blades. A less computationally expensive method is the actuator disc with rotation (AD+R) model, where the turbine blades are represented by a rotating circular disc that extracts energy from the wind through axial and azimuthal forces. The actuator disc without rotation (AD) is the simplest form, where only the axial force is considered. These methods have been applied in various LES models, such as the EllipSys3D model (Göçmen et al., 2016, AL), the MITRAS model (Salim et al., 2018, AD+R), the Simulator for Wind Farm Applications (SOWFA) (Fleming et al., 2014; Churchfield et al., 2016, AL), and the Parallelized Large-eddy simulation Model (PALM) (Witha et al., 2014; Maronga et al., 2015, 2020, AL, AD, AD+R). For applications that do not require a very high resolution and detailed near wake structures, the AD+R and AD methods are often favorable as they require fewer computational resources, are easier to implement, and yet achieve an adequate level of accuracy for far wake information (Breton et al., 2017)

Besides the accuracy of the turbine wake methods, a realistic ABL plays a crucial role in wake simulations. The processes that affect wind farms range from macro- and mesoscale weather phenomena, such as cyclones and fronts, down to the microscale of turbulence in the ABL (Porté-Agel et al., 2020). Given this multi-scale nature, a meso-to-micro numerical downscaling could be useful in simulating the ABL-turbine interaction realistically and understanding its underlying mechanisms (Muñoz-Esparza et al., 2014; Bakhoday-Paskyabi et al., 2022a). To incorporate realistic ABL conditions, one can use the meso-micro offline coupling approach, i.e. the output of a mesoscale model is used as the driven boundary condition for the dedicated LES models (e.g. Wang et al., 2020; Lin et al., 2021; Bakhoday-Paskyabi et al., 2022b; Onel and Tuncer, 2023). However, this approach depends on the frequency of the mesoscale model output, which is often not enough for short-time scale processes. The second approach, online coupling, uses a meso-micro coupled model system, where a mesoscale model and an LES model are coupled through a coupling interface. While the online coupling approach provides a seamless downscaling, it is more difficult to implement than the offline approach. The third approach, nested downscaling, is using a single model that can handle the downscaling naturally through a system of nested domains where the inner domains can be configured to run in LES mode. For example, the Consortium for Small-scale Modeling (COSMO) model (Baldauf et al., 2011) and the Weather Research and Forecast (WRF) model (Skamarock et al., 2019) provide this capability.

The WRF model is open-source software that is highly popular in the atmospheric research community for studying a wide range of atmospheric processes, from idealized studies to real-world applications. It facilitates a meso-to-micro nested downscaling framework capable of simulating turbine wakes under realistic Atmospheric Boundary Layer (ABL) conditions (Muñoz-Esparza et al., 2014; Bakhoday-Paskyabi et al., 2022a; Ning et al., 2023). There are several WRF implementations that include the effects of wind farms and wind turbines (e.g., Fitch et al., 2012; Volker et al., 2015; Mirocha et al., 2014; Kale et al., 2022), which can be grouped into wind farm parameterization (WFP) and wind turbine model (WTM).

The WFP approaches (Fitch et al., 2012; Volker et al., 2015) are commonly used to study the collective effects of wind turbine wakes on the ABL and the interactions between wind farms (e.g. Pryor et al., 2020; Fischereit et al., 2022). One advantage of these approaches is their low computational cost and ease of implementation. For example, (Fitch et al., 2012)

only requires the thrust and power curve data from turbine manufacturers. However, due to the limitation of the target horizontal resolutions, which must be at least 3 to 5 rotor diameters (Fischereit et al., 2022), the WFP cannot explicitly resolve individual turbine wakes or turbine-to-turbine interactions. This may result in inaccurate evaluations of wakes behind wind farms (Lee and Lundquist, 2017).

On the other hand, the implemented general actuator disc (GAD) model in WRF (Mirocha et al., 2014; Kale et al., 2022) employs the AD+R method based on the BEM theory. However, due to the requirement for high resolution, with at least a few grid points across the rotor disc, the simulation can become costly for large arrays of turbines or multiple wind farms. Additionally, the GAD model requires more information about the turbine, such as blade profiles and aerodynamic characteristics as well as generator speed and blade pitch control, which may sometimes be confidential and not publicly available. Finally, the GAD implementation's source code has not been released publicly, limiting its use for the scientific community, despite the WRF being open-source.

In this study, we developed a new wind turbine parameterization (WTP) model named the Simple Actuator Disc for Large Eddy Simulation (SADLES). SADLES bridges the gap between the Generalized Actuator Disc (GAD) and Wind Farm Parameterization (WFP) models within the Weather Research and Forecasting (WRF) model. It simulates turbine wakes explicitly at intermediate resolutions (e.g., tens of meters) between those achievable with GAD and WFP. Similar to the WFP model proposed by Fitch et al. (2012), SADLES employs the actuator disc (AD) method and requires only basic information about the turbines, such as their thrust and power curves. To accommodate the LES downscaling approach, we also implemented the cell perturbation method (Muñoz-Esparza et al., 2014), which is necessary for generating turbulence in the nested domains. The WRF-SADLES code package is an open-source software suite that includes both the SADLES module and the cell perturbation module. Its development aims for integration into the official WRF repository, thereby promoting further open research within the weather forecasting community.

To validate the WRF-SADLES model, we chose to compare its idealized simulations of a 5-MW wind turbine with similar simulations using the PALM model. The reason for selecting PALM was that it includes a wind turbine model that uses the AD+R method, which is comparable with the AD method used by the WRF-SADLES, but with higher accuracy. Moreover, the PALM model has been shown to agree well with more sophisticated wake models and observations (Witha et al., 2014; Vollmer et al., 2015; Bakhoday-Paskyabi et al., 2022a, b). In addition, we demonstrated a more realistic application of the WRF-SADLES by downscaling the ERA5 reanalysis data (Hersbach et al., 2020) to a 40-meter resolution around the Alpha Ventus wind parks, enabling us to investigate the effects of turbine-to-turbine and farm-to-farm interactions. The simulations are then compared with the observational data recorded at the FINO1 offshore meteorological mast station.

The rest of the paper is organized as follows: Section 2 describes the the method and implementation of WRF-SADLES. Section 3 presents an idealized simulation with a single 5-MW wind turbine and compares it with the PALM model. Section 4 offers an example application of a multi-scale downscaling approach with multiple wind farm simulations. Finally, Section 5 discusses the potential applications and limitations of the SADLES model.

## 2 Methods

### 2.1 The simple actuator disc for large eddy simuation

The actuator disc (Anderson, 2020) is a hypothetical surface perpendicular to the wind flow that extracts energy continuously from the ambient wind through the work of the thrust force:

$$\mathbf{F}_T = -\frac{1}{2}\rho C_T |\mathbf{V_0}|\mathbf{V_0} A, \tag{1}$$

where $\rho$ is the air density, $A$ is the rotor area, $C_T$ is the thrust coefficient, which is a function of the ambient (unperturbed) wind speed $|\mathbf{V_0}| = \sqrt{u_0^2 + v_0^2}$. The wind speed at the rotor, $\mathbf{V}$, is reduced by the axial induction factor $a$ through the relation:

$$\mathbf{V} = \mathbf{V_0}(1-a). \tag{2}$$

The tendency terms of the thrust force are incorporated into the WRF model at the grid cells where the actuator disc intersects:

$$\frac{\partial u}{\partial t}\bigg|_T = -\frac{1}{2}C_T |\mathbf{V_0}|u_0 \frac{\delta A}{\Delta x \Delta y \Delta z}, \tag{3}$$

$$\frac{\partial v}{\partial t}\bigg|_T = -\frac{1}{2}C_T |\mathbf{V_0}|v_0 \frac{\delta A}{\Delta x \Delta y \Delta z}, \tag{4}$$

where $\delta A$ is the portion of the actuator disc area within the grid cell, $\Delta x$, $\Delta y$, and $\Delta z$ are the grid sizes. We can shorten the formula by defining the area factor $F_A = \delta A/(\Delta x \Delta y \Delta z)$, which can be determined by performing a vertical and horizontal discretization of the actuator disc area (Fig. 1).

We note that in the WRF model, the tendency terms are defined for the coupled velocity, which is defined as $U = \mu_d u$, with $\mu_d$ as the dry mass of the air column. It is also more convenient to use the wind speed at the rotor disc instead of the ambient wind speed. Thus the tendency terms to be added to the model become:

$$\frac{\partial U}{\partial t}\bigg|_T = -\frac{1}{2(1-a)^2}\mu_d C_T |\mathbf{V}|u F_A \tag{5}$$

$$\frac{\partial V}{\partial t}\bigg|_T = -\frac{1}{2(1-a)^2}\mu_d C_T |\mathbf{V}|v F_A \tag{6}$$

In the wind farm parameterization proposed by Fitch et al. (2012), turbulent kinetic energy (TKE) is introduced in proportion to the difference between the thrust coefficient and the power coefficient ($C_T - C_P$). This addition is justified by the extraction of kinetic energy from the mean flow, where part contributes to power production, and the remainder is transferred to TKE. The question arises: is this approach valid or necessary for the micro-scale actuator disc model? For instance, the exclusion

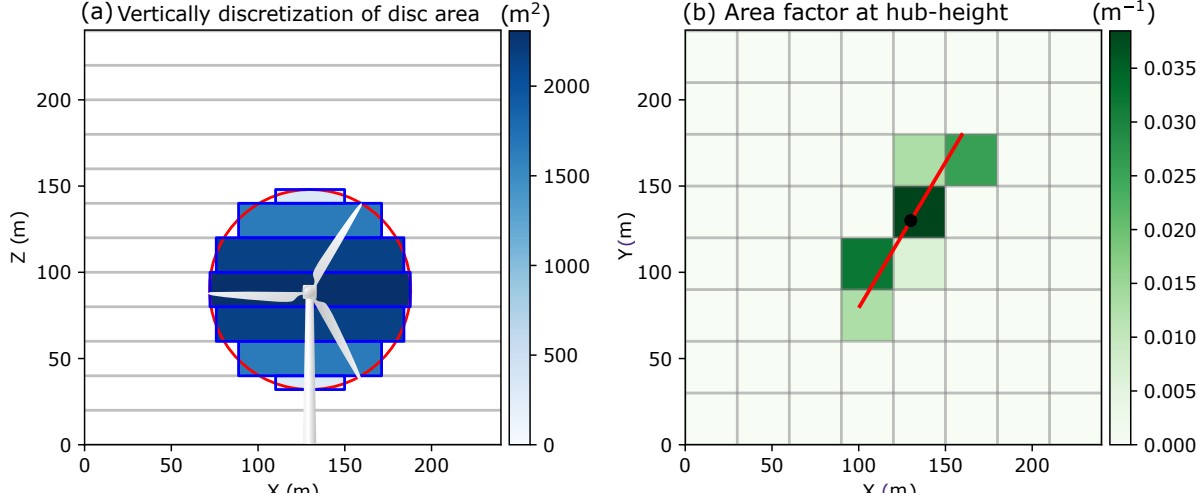

**Figure 1.** The illustration demonstrates the discretization of an actuator disc. (a) Vertically, the disc is divided into sections between each full vertical level. (b) The discretized area factor at the hub-height level is depicted. The actuation disc radius is 116 meters, and both horizontal and vertical grid sizes are 30 meters for illustrative purposes.

of rotational effects can be viewed as subgrid-scale turbulence impacting wake recovery. To investigate, we adopt the method used by Fitch et al. (2012), incorporating a source of subgrid-scale TKE as:

$$\left.\frac{\partial TKE_{sgs}}{\partial t}\right|_T = \frac{1}{2(1-a)^3}\mu_d C_{TKE}|\mathbf{V}|^3 F_A \tag{7}$$

where $C_{TKE} = f_{TKE}(C_T - C_P)$, $C_P$ is the power coefficient, and $f_{TKE}$ is a factor that controls the amount of kinetic energy loss is converted to TKE.

The tendency terms above depend critically on the axial induction factor $a$. Some previous studies have used certain specific values of $a$, such as $a = 1/4$ (Calaf et al., 2010) or even $a = 0$ (Fitch et al., 2012) (i.e. they used the wind speed at the grid point directly instead of the unperturbed wind speed). In our implemented SADLES model, we provide two options for estimating $a$:

- Option 1 (direct evaluation): First, the hub-height ambient wind speed $|\mathbf{V_0}|$ is evaluated at two rotor diameters ($2D$) in
front of the turbine. Then, the induction factor is calculated by:

$$a = 1 - \frac{|\mathbf{V}|}{|\mathbf{V_0}|},$$

- Option 2 (inferred evaluation): In this option, only the hub-height wind speed at the rotor location is needed. Instead, we assume the 1-dimensional momentum theory ($C_T = 4a(1-a)$) and therefore:

$$a = \tfrac{1}{2}(1 - \sqrt{1 - C_T(|\mathbf{V}|)}).$$

Note that although the thrust curve is typically provided as a relation between $C_T$ and the ambient wind speed $|\mathbf{V}_0|$, we can also establish the relation between $C_T$ and the wind speed at the rotor $|\mathbf{V}|$ using the 1-dimensional momentum theory.

In general, the direct evaluation of $a$ (Option 1) is more intuitive. However, this method has some potential problems, including the formula being affected by the blockage effect, where the wind speed in front of the wind turbine is reduced. By

using this formula, $a$ can exceed 0.5, implying that the wind behind the turbine becomes opposite to the ambient wind. This is nonphysical and can lead to model instability. Thus, the upper limit of $a$ is set to the inferred evaluations of $a$ (Option 2).

## 2.2 Cell perturbation

Traditional LES simulations often use periodic lateral boundary conditions, which allow turbulent eddies to fully develop into a pseudo-equilibrium state. However, in our LES downscaling approach, the limited time and space available at the inflow

boundary can prevent eddies from fully developing, particularly in cases with a small inner domain, high ambient wind speed, or stable boundary layer conditions. This can lead to incomplete development of the eddies and potentially affect the accuracy of the simulation.

The above problem can be alleviated using the cell perturbation method (Muñoz-Esparza et al., 2014, 2015), which is a simple and effective way to improve the realism of turbulent representations. The method adds a random perturbation of

potential temperature within the interval [–0.5, 0.5] to three cells of $8\times8$ grid points near the chosen boundaries. In the idealized setup of Muñoz-Esparza et al. (2014), perturbations are introduced at every vertical grid point up to two-thirds of the inversion layer, which is known in the idealized setting. In our approach, the perturbations are applied with full magnitude up to a predefined vertical level $k_1$ and then gradually reduced to zero at a higher level $k_2$ using the weight $\cos^2\left[0.5\pi(k - k_1)/(k_2 - k_1)\right]$, where $k$ is the vertical level. As noted by Muñoz-Esparza et al. (2014), the perturbation process should not be done at every

time step, but at an interval that is at least equal to the perturbation time scale, $T_s = 8\Delta x/U$, where $U$ is the characteristic velocity scale and $\Delta x$ is the horizontal grid spacing.

Because the cell perturbation method is not available with the distribution of WRF, we included our implementation within the distribution of WRF-SADLES.

## 2.3 Code implementation and turbine information

We've integrated the SADLES code into the Advanced Research WRF (ARW), version 4.3.1, with MPI support. This was achieved by adding two new Fortran 90 modules: a SADLES module (*module_sadles.F*) and a cell perturbation module (*module_cellpert.F*), both serves as the WRF's physics package (located in the *phys* directory). The new namelist options for these modules were incorporated into the WRF model by modifying the WRF's Registry. During the first step of the Runge-Kutte loop (within *module_first_rk_step_part1.F*), tendency terms of the SADLES module are added to the model, while the potential

temperature perturbation from the cell perturbation is incorporated after the integration loop (within *solve_em.F*).

**Table 1.** Domain configurations for the idealized WRF (prefix W) and PALM (prefix P) experiments

| Experiments | Domain | $N_x \times N_y \times N_z$ | $\Delta x [m]$ | $\Delta t$ [s] | $L_x$ [m] | $L_y$ [m] |
|---|---|---|---|---|---|---|
| W30m_Opt1, W30m_Opt2, | D01 | $322 \times 163 \times 91$ | 90 | 1/2 | 28890 | 14490 |
| W30m_TKE0, W30m_TKE1 | D02 | $322 \times 163 \times 91$ | 30 | 1/6 | 9660 | 4860 |
| W10m_Opt1, W10m_Opt2, | D01 | $322 \times 163 \times 91$ | 30 | 3/20 | 9660 | 4860 |
| W10m_TKE0, W10m_TKE1 | D02 | $448 \times 163 \times 91$ | 10 | 1/20 | 4470 | 1620 |
| P30m | D01 | $336 \times 172 \times 90$ | 90 | 1/2 | 30240 | 15480 |
| | D02 | $336 \times 192 \times 160$ | 30 | 1/6 | 10080 | 5760 |
| P10m | D01 | $336 \times 176 \times 160$ | 30 | 3/20 | 10080 | 5280 |
| | D02 | $432 \times 192 \times 150$ | 10 | 1/20 | 4320 | 1920 |

Tables A1 and A2 outline the available options in the SADLES and cell perturbation modules. This modified WRF system also enables the simulation of turbine behavior in idealized experiments, with additional namelist options for specifying the Coriolis parameter and surface roughness length. The code has been released as open-source to foster open research (Bui, 2023), with the aspiration of its inclusion in the official WRF repository. Users can implement the WRF-SADLES by copying these modules and overriding a few related files within the existing WRF file structure, followed by recompiling the model system. In the subsequent section, we utilize turbine data from Larsén and Fischereit (2021), which provides the locations, thrust curves, and power curves of wind turbines from various wind farms in the North Sea.

## 3 Idealized simulation

### 3.1 Experiment design

To start, we investigate the calculation of the axial induction factor (e.g., Option 1 or Option 2) and the additional tendency for sub-grid TKE (i.e., $f_{TKE}$). Due to challenges in acquiring relevant observational data, we conducted and compared idealized experiments employing a single 5-MW turbine using the WRF-SADLES and PALM models, known for their realistic simulation of turbine wakes. Simulations were executed at two resolutions: 10 meters (high resolution) and 30 meters (target intermediate resolution).

Table 1 summarizes the experiments conducted using the WRF-SADLES and PALM models. For the WRF-SADLES simulations, experiments with suffixes _Opt1 and _Opt2 represent different options for calculating the axial induction factor $a$, with Option 1 employing direct evaluation and Option 2 employing inferred evaluation. In these experiments, $f_{TKE}$ takes the default value of 0.5. Additionally, experiments with suffixes _TK0 and _TKE1, both utilizing Option 2, aim to investigate the effect of adding TKE tendency as in Equation (7), with $f_{TKE}$ set to 0 and 1, respectively.

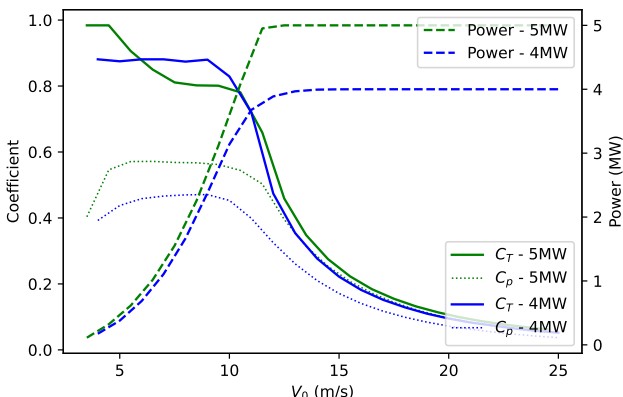

**Figure 2.** Thrust coefficient ($C_T$), power coefficient ($C_P$), and turbine power as functions of ambient wind speed ($V_0$) are shown for the 4 MW and 5 MW wind turbines used in the idealized (5 MW) and realistic (4 MW and 5 MW) experiments in this paper.

The domain configurations for each experiment are summarized in Table 1. Each experiment utilized two nested domains (D01 and D02), with the outer domain D01 having a coarser resolution (30 meters or 90 meters) and using periodic boundary conditions on all sides, and the inner domain D02 (10 meters or 30 meters) applying cell perturbation at the inflow boundary on the west side. All domains except the 10-meter domain had an aspect ratio of 2:1. The 10-meter domain had a longer aspect
ratio of 2.76:1 to allow the turbulence and turbine wake to evolve over a longer distance.

The top model level for all experiments is set at 1600 meters, with an 800-meter Rayleigh damping layer at the top with a coefficient of $0.2 \text{ s}^{-1}$. There is no vertical level stretching for the 30-meter resolution experiments. In the case of the 10-meter experiments, the vertical levels are stretched such that the near-surface vertical resolution is roughly 10 m. The initial potential temperature is 288 K from the surface up to 500 m, and then increases with a lapse rate of 1 K/100 m. We configured an
idealized weak convective boundary layer with a surface turbulence heat flux of $\overline{(\theta' w')}_s = 0.02 \text{ K m s}^{-1}$, similar to some previous studies (Muñoz-Esparza et al., 2014; Kale et al., 2022). After a spin-up time of 20 hours, which is performed for the outer domain only, a quasi-equilibrium, well-mixed boundary layer is established. In order to have the wind in the boundary layer roughly in the zonal direction, the geostrophic wind is set to rotate slightly to the left, specifically, $U_g = 10.24 \text{ m/s s}^{-1}$ and $V_g = -1.39 \text{ m s}^{-1}$. After the spin-up time, both domains are integrated for 4 hours with a 1-minute output interval for the
inner domain (D02) for further analysis.

In our idealized experiments, no moisture is initialized. Except for the surface layer, all other physical parameterization schemes, including microphysics, cumulus, boundary layer, and radiation, are turned off. For the surface layer parameterization, we used the Revised MM5 Monin-Obukhov surface layer scheme (Jiménez et al., 2012). Similar to the idealized experiment, to enable the LES mode, we used the 1.5-order three-dimensional LES turbulence closure, in which the subgrid-scale TKE

is treated as a prognostic variable (Lilly, 1967). Other settings include: the Coriolis parameters are set to $1.177 \times 10^{-4} \text{s}^{-1}$ (54°N), and the surface roughness length is set to 1 mm.

At the center of the inner domain, we placed a 5 MW wind turbine used at the Alpha Ventus wind farm. The turbine information taken from Larsén and Fischereit (2021) includes a rotor diameter of 116 m, a hub height of 90 m, and the thrust and power curves at different wind speeds (Fig. 2).

**PALM configurations**

To evaluate the performance of the SADLES module in simulating turbine wakes, we compared the results from the WRF-SADLES model with those from the PALM model, maronga2015parallelized,maronga2020overview, system 21.10 revision r4901. PALM is an LES model developed at Leibniz Universität Hannover, Germany, and has been shown in several studies to be capable of simulating wind turbine wakes effectively (e.g. Witha et al., 2014; Vollmer et al., 2015).

In PALM, the wind turbine is represented by an advanced actuator disc with rotation, which calculates both the thrust and torque forces as functions of radius and tangential angle from the center of the rotor. Similar to the GAD methods (Mirocha et al., 2014; Kale et al., 2022), the wind turbine model in PALM computes the local lift and drag forces based on the BEM method, which is accurate but requires additional information on the turbine and blade aerodynamic properties. For this reason, currently only three types of wind turbines are officially supported, including the National Renewable Energy Laboratory

(NREL) 2.3-MW, 5-MW and 15-MW models (Jonkman et al., 2009; Gaertner et al., 2020; Ardillon et al., 2023). To compare with WRF-SADLES, we used the NREL 5-MW model with the same hub height of 90 m. However, the rotor diameter of the NREL 5-MW is slightly larger at 128 meters compared to the 116-meter diameter of the 5-MW turbine used in WRF-SADLES.

The two idealized experiments, P10m and P30m, with two nested domains similar to the WRF-SADLES experiments (see Table 1 for domains configurations). Cyclic lateral boundary conditions were used for the coarser domain. To prepare initial

conditions for the main run we used a precursor run. Precursor domains are very smaller than main simulations. It has been defined $96 \times 64$ grid points for both 30m and 10m simulations in precursor run. Number of vertical levels for each precursor run is the same as the main run. The model has implemented for 86400 seconds to reach a steady state condition in precursor mode. To parameterize subgrid-scale (SGS) turbulence in PALM we used 1.5-order closure (according to Deardorff (1980) and modified by Moeng and Wyngaard (1988) and Saiki et al. (2000)). We considered atmospheric conditions similar to those

in the WRF model, including: an initial potential temperature of 288 K, a surface turbulence heat flux of $\overline{(\theta' w')_s} = 0.02$ W m$^{-2}$, and a geostrophic wind of approximately 10 m/s from the west.

### 3.2 Result

Figure 3 shows the average wind speeds over 4 hours for the four WRF-SADLES idealized experiments, varying in resolutions and axial induction options. Noticeably, the averaged wake angles, reflecting average wind direction, differ between the 10-

meter and 30-meter resolutions, suggesting a dependency of momentum fluxes on model solutions. Furthermore, the 10-meter resolution experiments exhibit slower wake recovery and a smaller rate of wake expansion, indicating potentially stronger turbulence activities at lower resolutions. Conversely, minimal difference is observed between the two options for calculating

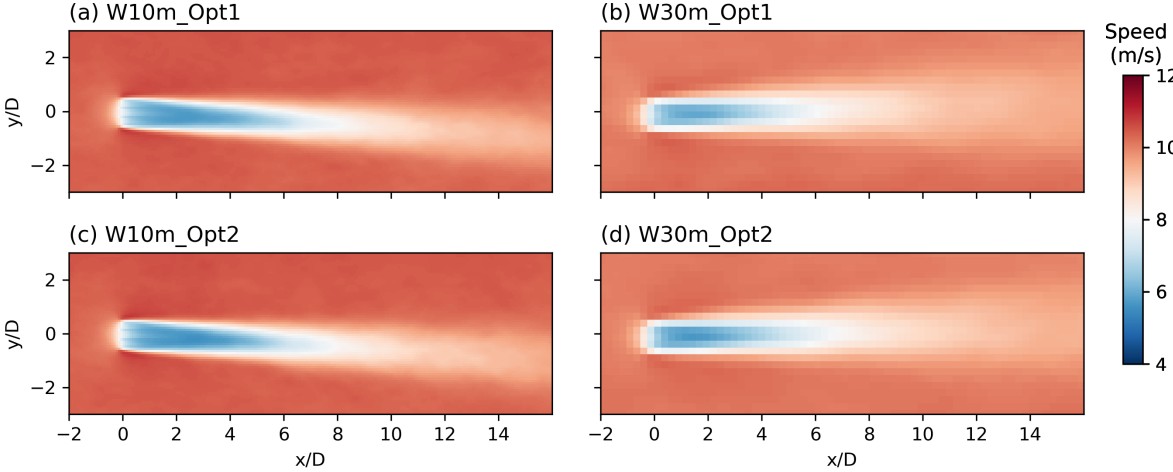

**Figure 3.** 4-hour average wind speeds at the turbine hub height (90 m) from idealized WRF-SADLES simulations with different resolutions and options for axial induction factors.

the axial induction factor, with visually identical averaged wind speeds. This validates the 1-D momentum theory of the actuator disc, which is used for calculating axial induction factor $a$. Given that Option 1 necessitates wind speed evaluation in front

of the turbine, it may be sensitive to model resolution and blockage effects. Therefore, we recommend the use of Option 2 in practical applications. Subsequent sections will explore the effect of the added TKE source and compare WRF-SADLES simulations with PALM simulations.

Figure 4 depicts a comparison of the averaged wind speeds between PALM simulations and WRF-SADLES with $f_{TKE} = 0$ or 1. Interestingly, for the same resolution, minimal variation is observed in the WRF-SADLES experiment, indicating that

the wake may not be significantly affected by the added subgrid-scale TKE at the actuator disk. However, a more noticeable difference is evident between WRF-SADLES and PALM. This disparity could be attributed, at least in part, to methodological differences (e.g., simple actuator disk versus actuator disk with rotation) and potentially differing turbine properties.

Both the PALM and WRF-SADLES simulations show similar wake shapes at the 10-meter resolution (Fig. 4 a, c, e). However, near the turbine, PALM predicts a lower wind speed of around 4 m s$^{-1}$ compared to WRF-SADLES' 6 m s$^{-1}$. Ad-

ditionally, the wake in the PALM simulation expands slightly with increasing distance from the turbine, while WRF-SADLES exhibits minimal expansion. This difference might be attributed to the absence of the rotational effect in WRF-SADLES, which is included in PALM.

At the 30-meter resolution (Fig. 4b, d, f), WRF-SADLES shows consistent wake intensity and shape compared to the 10-meter results. However, the wake in PALM experiment at 30-meter resolution (Fig. 4b) is much weaker and narrower than in

at 10-meter resolution. The cause of this discrepancy is unclear, but one possibility is related to the interpolation between the actuator disc polar grid system and the Cartesian coordinate system used in the PALM model.

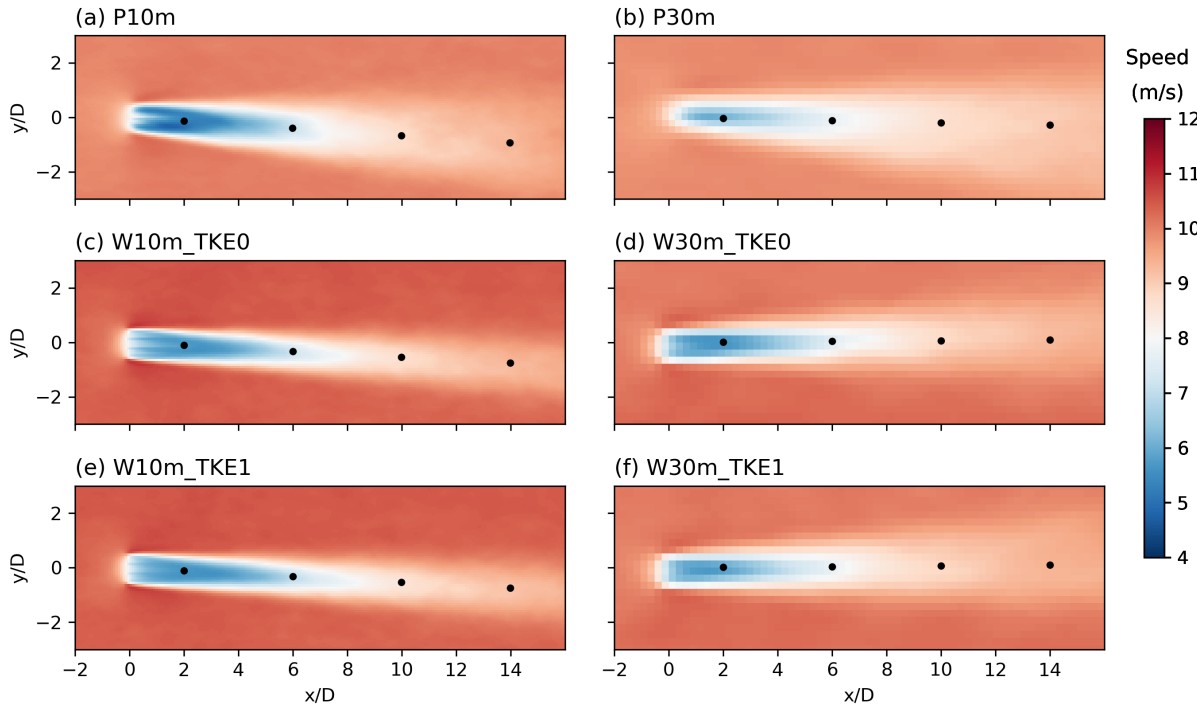

**Figure 4.** Similar to Fig. 3, but for PALM simulations and WRF-SADLES simulations with different $f_{TKE}$ values (=0 for _TKE0, =1 for _TKE1). Black dots indicate distances of $2D$, $6D$, $10D$, and $14D$ behind the turbine.

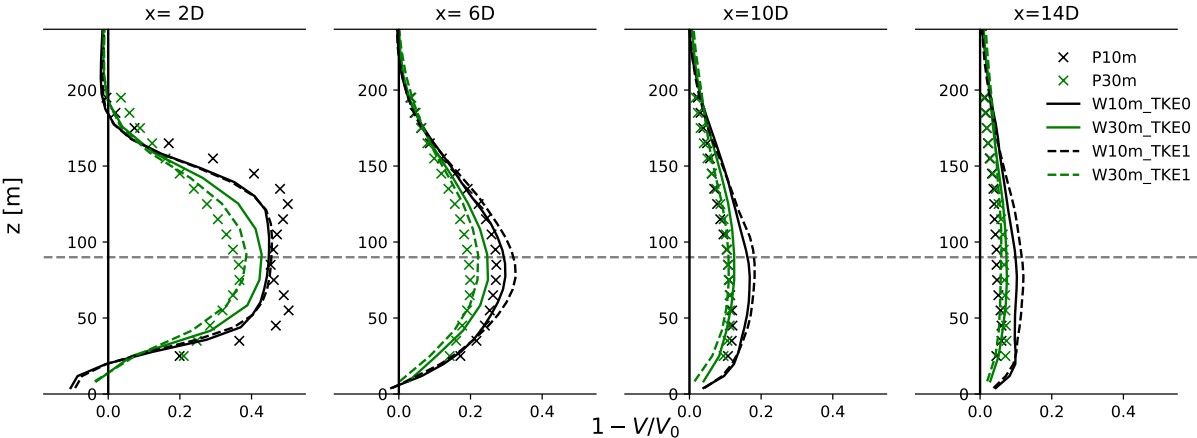

**Figure 5.** Vertical wake deficit $(1 - V/V_0)$ for idealized experiments at distances behind the turbine of two turbine diameters $(2D)$, $6D$, $10D$, and $14D$. The evaluation points are indicated by black dots in Fig. 4.

We calculated the vertical wake deficit at distances behind the turbine of two turbine diameters ($2D$), $6D$, $10D$, and $14D$ (as shown in Fig. 5). At the near-wake distance of $2D$, the wake deficits agree at around 50% for both PALM and WRF-SADLES. However, the P10m deficit exhibits a distinct shape with two peaks, one above and one below the hub height, due to the nature of the BEM method. At 30 meters, the wake deficit is weaker for both models, though the W30m_TKE0 experiment shows the closest agreement to the higher-resolution simulations.

As the distance from the turbine increases, both the PALM and WRF-SADLES models show similar wake deficit profiles with a single peak located slightly below the hub height. However, their wake recovery rates differ. In the PALM simulations, the 10-meter resolution wake recovers faster than the 30-meter one, resulting in a larger deficit at 14D for the P30m simulation. Conversely, the WRF-SADLES wakes recover slower at 10 meters, leading to consistently higher deficits compared to the 30-meter simulations. Interestingly, using the P10m simulation as a reference, the WRF-SADLES simulation without added subgrid-scale turbulence (W10_TKE0) shows better agreement than the one with added TKE (W10_TKE1). While W10_TKE0 aligns better at near-wake distances, the 30-meter simulation without added TKE (W30_TKE0) exhibits better agreement at far-wake distances ($10D$ and $14D$).

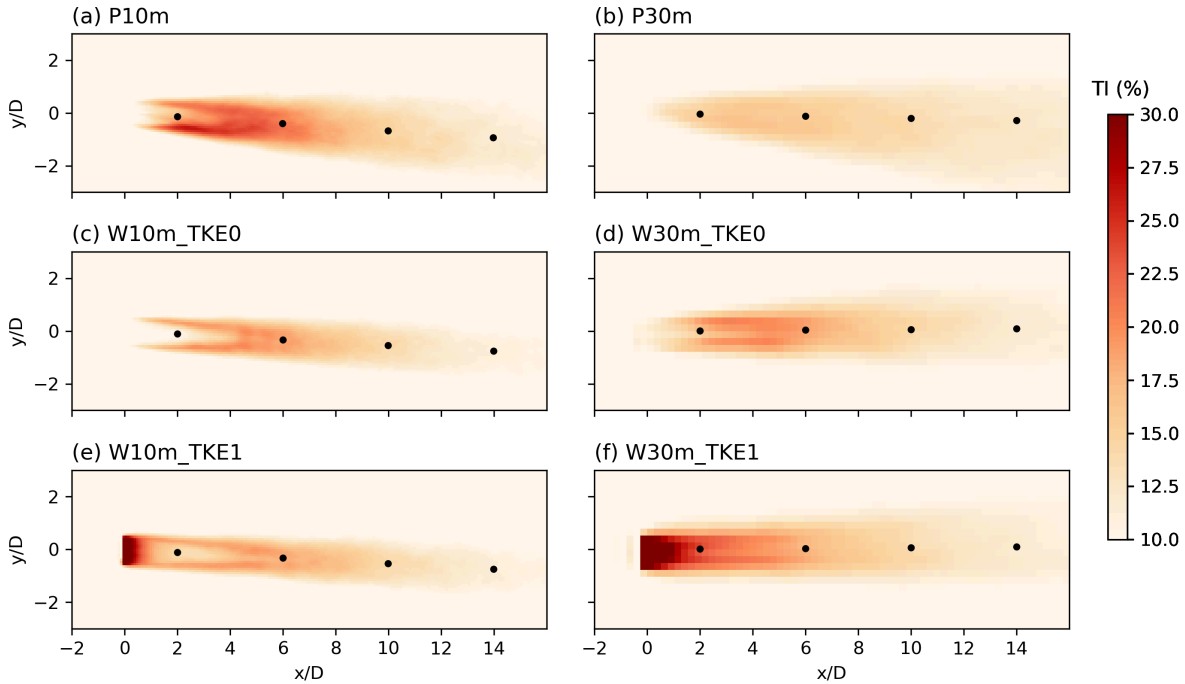

**Figure 6.** Similar to Fig. 4, but for the time-averaged turbulence intensity $\overline{TI}$.

To gain insight into why the added subgrid-scale TKE source has minimal impact on the wake in WRF-SADLES, we compared the calculated average turbulence intensity (TI) between the experiments. The average TI is computed using the equation:

$$\overline{TI} = \frac{\sqrt{\frac{2}{3}\overline{TKE}}}{|\mathbf{V}|}, \tag{8}$$

where $\overline{TKE}$ represents the time-averaged total turbulence kinetic energy, is the sum of gridscale ($\overline{TKE}_{gs}$) and subgrid-scale

($\overline{TKE}_{sgs}$) terms. $\overline{TKE}_{sgs}$ is a prognostic variable derived from the 1.5-order turbulence closure within the PALM and WRF-SADLES models. On the other hand, the gridscale term is derived from $\overline{TKE}_{gs} = \frac{1}{2}(\overline{u'^2} + \overline{v'^2} + \overline{w'^2})$, where $u', v', w'$ denote the deviations of wind speed components from their respective time averages over the simulation period.

Figure 6 compares the average TI at the hub height for the PALM and WRF-SADLES simulations, including cases with and without added subgrid-scale TKE sources in WRF-SADLES. Outside the wake regions, all experiments show a consistent TI of

around 10%. Both the PALM and WRF-SADLES simulations exhibit TI development at the turbine edges due to shear production. The maximum TI reaches approximately 20% at distances beyond $2D$. In contrast, the WRF-SADLES experiment with added subgrid-scale TKE (_TKE1) achieves the intended effect, with a maximum TI exceeding 30% at the turbine location. However, this additional subgrid turbulence quickly dissipates as it advects downstream. Within about $2D$ for W10m_TKE1 and $4D$ for W30m_TKE1, the TI levels in the simulations with added subgrid-scale TKE become comparable to those without.

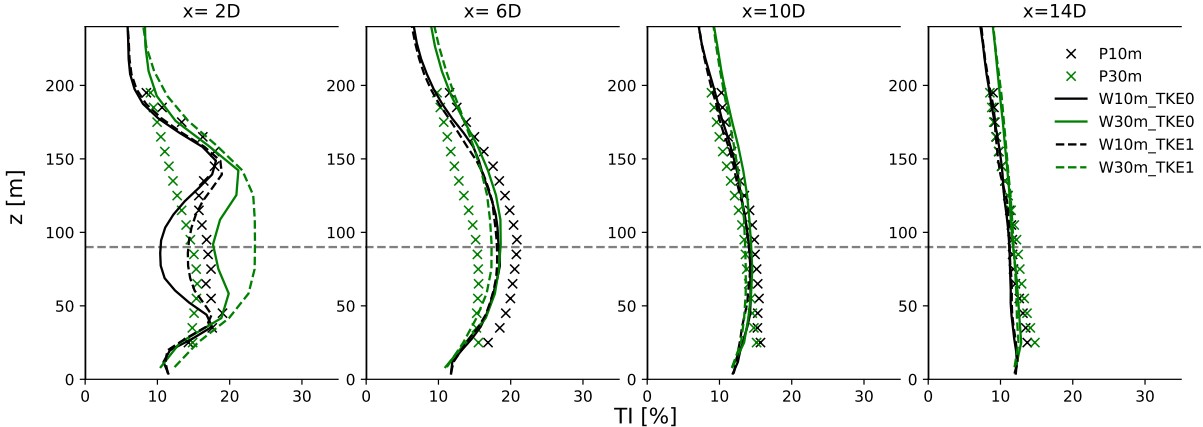

**Figure 7.** Similar to Fig. 5 but for the time averaged turbulence intensity $\overline{TI}$.

Figure 7 presents the vertical structure of the time-averaged turbulent intensity (TI) at various distances behind the wind turbine. Significant discrepancies in the TI are observed between the two models (PALM and WRF-SADLES) and resolutions (10 meters and 30 meters) at the near-wake distance of $2D$. The experiments with 10-meter resolution show two TI peaks above and below the hub height, likely due to the shear production of turbulence. These peaks reach a TI of about 20% and are consistent between the PALM and WRF-SADLES models. The WRF-SADLES model with added subgrid-scale TKE shows

a TI about 5% higher than the non-added TKE counterpart. The W30m_TKE1 experiment has the highest TI value, reaching approximately 25%.

Beyond $6D$ downstream, both resolutions of the WRF-SADLES simulations show consistent TI predictions. However, discrepancies remain between the 10-meter and 30-meter resolutions in the PALM simulations. The 30-meter PALM experiment exhibits a significantly lower TI compared to all others, while the 10-meter version shows the highest TI. As the wake recovers further downstream, the TI variations across all simulations weaken. This weakening of wake turbulence brings them closer to background levels, typically around 10% near the hub height and decreasing with height. This convergence typically occurs by 14D downstream.

Using the 10-meter PALM simulation as a reference, the WRF-SADLES model displays a similar TI structure, especially at far-wake distances. Adding a tendency term for subgrid-scale TKE in WRF-SADLES primarily impacts the near-wake region and has minimal effect further downstream. For both near and far wakes, the experiments without added subgrid-scale TKE in WRF-SADLES show better agreement with the reference experiment for both resolutions (30-meter and 10-meter). Therefore, we recommend using the WRF-SADLES model without the added subgrid-scale TKE source (i.e., $f_{TKE} = 0$) in practical applications.

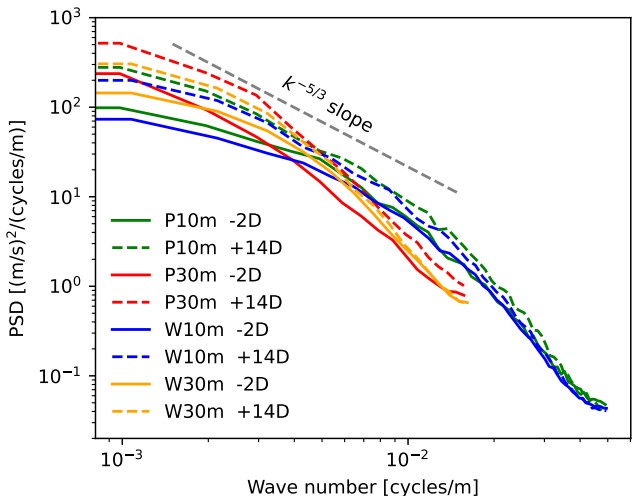

**Figure 8.** Time-averaged power spectral densities (PSDs) of wind speed spatial fluctuations for the PALM (P10m, P30m) and WRF-SADLES (W10m, W30m) simulations at two locations: $-2D$ upstream and $+14D$ downstream of the wind turbine. The analysis was computed from a meridional cross-section at hub height with a length of $8D$, centered on the wake axis.

To further assess the turbine's influence on turbulence, we analyzed the time-averaged wavenumber power spectral densities (PSDs) at $2D$ upstream and $14D$ downstream, revealing the turbulence characteristics at pre-wake and post-wake distances, respectively (Fig. 8). For consistent comparisons across simulations, the wavenumber spectral analysis was conducted in a meridional cross-section at hub height with a length of $8D$. To improve clarity and avoid redundancy, we present the average results of all WRF-SADLES simulations at each resolution (W30m and W10m for 30-meter and 10-meter resolutions, respectively) since their turbulence properties exhibit minimal variation across different options within the model.

Both WRF-SADLES and PALM simulations exhibit similar trends in turbulence properties. Simulations with a 10-meter resolution capture higher turbulence levels, particularly for smaller-scale structures (wavenumbers above $4 \times 10^{-3}$ cycles/m, corresponding to wavelengths shorter than 250 meters). This indicates a better ability to resolve these structures with finer resolution.

The 30-meter simulations show a faster dissipation of turbulence energy compared to the Kolmogorov power law (represented by a -5/3 slope) at smaller scales (wavenumbers above $4 \times 10^{-3}$ cycles/m). This suggests limitations in capturing the energy transfer mechanisms at these scales. In contrast, the 10-meter simulations exhibit a broader range of applicability for the Kolmogorov power law, spanning from approximately $2 \times 10^{-3}$ to $10^{-3}$ cycles/m (corresponding to wavelengths between 100 meters and 500 meters). This wider range signifies a more accurate representation of the energy cascade across different scales in these simulations.

Furthermore, all simulations show an increase in turbulence energy at the far-wake distance ($14D$) compared to upstream turbulence ($-2D$), for all wavenumbers. This increase is more pronounced for lower wavenumbers (longer wavelengths) than for higher wavenumbers. This behavior is consistent for both PALM and WRF-SADLES models.

## 4 Meso-to-micro realistic downscaling simulation

This section demonstrates meso-to-micro downscaling using global reanalysis data. We employ the ERA5 data set from the European Centre for Medium-Range Weather Forecasts (ECMWF) (Hersbach et al., 2020). This data has a spatial resolution of approximately 31 kilometers and is available hourly. First, we briefly compare the simulation results with observational data from the FINO1 meteorological station in the southern North Sea. This comparison aims to assess the model's ability to reproduce real-world conditions. Subsequently, we will provide an example illustrating power loss due to farm-to-farm interaction.

### 4.1 Model configurations

To achieve turbine-scale resolution, we employed a system of five nested domains (detailed in Table 2). Each domain progressively reduces its grid size for finer resolution. The first three domains (D01, D02, D03; see Fig. 9a) focus on downscaling the mesoscale processes, while the final two domains (D04, D05; see Fig. 9b) transition to Large Eddy Simulation (LES) for high-resolution wind flow near the turbines.

The outermost domain (D01) has a resolution of 9 kilometers, encompassing a vast region that includes Europe and the North-East Atlantic Ocean (Figure 9a). The second domain (D02) zooms in on the North Sea (Figure 9a). The remaining three domains are centered around the Alpha Ventus wind farm, located near the FINO1 meteorological mast station (Figure 9b).

The first LES domain (D04) has a resolution of 200 meters and acts as an intermediate step between the meso-scale and turbine-scale domains. It does not include wind turbines within its simulation. The innermost domain (D05) has the highest resolution of 40 meters and encompasses a smaller area (19.2 km $\times$ 19.2 km) compared to a single grid cell of the original ERA5 data. This is the domain where the SADLES model is activated for simulating turbine wakes.

**Table 2.** WRF domain configurations for the real-data downscaling experiments.

| Domain | $N_x \times N_y \times N_z$ | $\Delta x$ (m) | $\Delta t$ [s] | $L_x$ [km] | $L_y$ [km] |
|--------|------------------------------|----------------|-----------------|-------------|-------------|
| D01 | $385 \times 321 \times 60$ | 9000 | 45 | 3456 | 2880 |
| D02 | $481 \times 382 \times 60$ | 3000 | 15 | 1440 | 1143 |
| D03 | $322 \times 322 \times 60$ | 1000 | 5 | 321 | 321 |
| D04 | $321 \times 321 \times 60$ | 200 | 1 | 64 | 64 |
| D05 | $481 \times 481 \times 60$ | 40 | 1/5 | 19.2 | 19.2 |

In domain D05, four wind farms are present comprising 5-MW and 4-MW turbines. Notably, the Alpha Ventus wind farm, featuring twelve turbines, is situated to the right of the FINO1 mast station. Detailed turbine specifications, including power and thrust curves, can be referenced in Larsén and Fischereit (2021) (refer to Fig. 2). For the subsequent experiments, we adopted the SADLES Option 2 for the inferred evaluation of the axial induction factor, along with the $f_{TKE}$ value of 0.

For vertical grid resolution, we adopted 60 levels following Bui and Bakhoday-Paskyabi (2022). This setup provides high resolution near the surface, approximately 10 meters, with 21 levels below 500 meters height. Regarding physical parameterization, we employed the following options: the RRTMG (Rapid Radiative Transfer Model for General Circulation Models) scheme (Iacono et al., 2008) for radiation parameterization across all domains, the Thompson graupel scheme (Thompson et al., 2008) for microphysics parameterization. The Tiedtke scheme (Zhang et al., 2011) was utilized for cumulus parameterization in the outermost 9-km domain (D01), with cumulus parameterization disabled in finer resolution domains. For surface layer parameterization, we utilized the Monin-Obukhov Similarity scheme (Jiménez et al., 2012), and the Noah Land Surface Model (Mukul Tewari et al., 2004) was employed for land surface parameterization in all domains. The Yonsei University (YSU) scheme (Hong et al., 2006) was chosen for PBL parameterization in mesoscale domains (D01-D03), while PBL parameterization was disabled in LES (Large Eddy Simulation) domains (D04, D05). To simulate large eddies in LES domains, we employed the 1.5-order three-dimensional TKE closure (Lilly, 1967), supplemented by the cell perturbation method along the southern and western boundaries for the first 16 levels from the surface (up to approximately 300-meter height) to initiate turbulence development.

## 4.2 Comparison with observational data

This section briefly evaluates the performance of WRF-SADLES using observations from the FINO1 meteorological mast station. We compare wind speed and direction data collected by cup anemometers at 90 meters (hereafter referred to as "mast data") with WRF-SADLES simulations. Additionally, we utilize data from a Windcube© 100s wind LiDAR (Kumer et al., 2014) (hereafter referred to as "LiDAR data"). The LiDAR data provides vertical profiles of wind speed and direction from vertical scans, along with radial wind speed from horizontal scans.

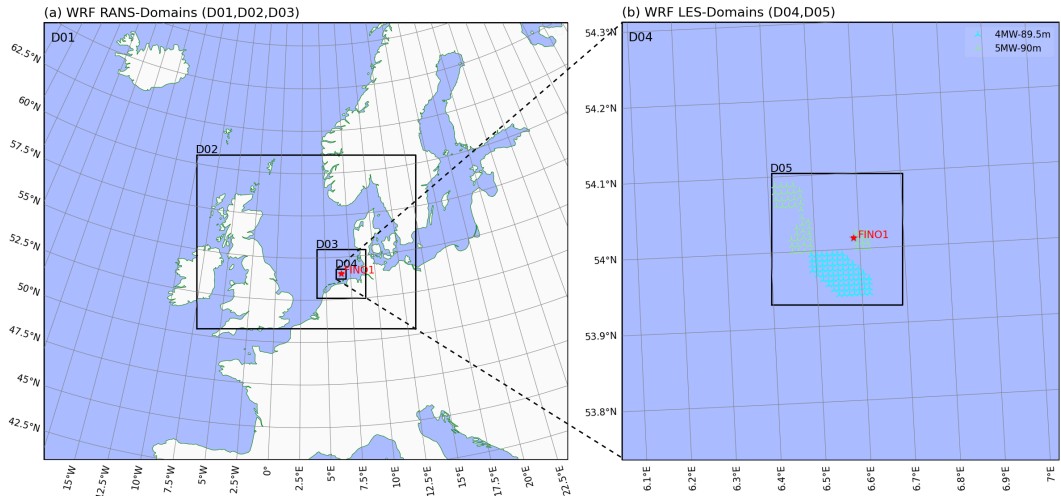

**Figure 9.** WRF domains used in the meso-to-micro downscaling simulations. The first three domains (a, D01-D03) are for mesoscale simulations, while the two innermost domains (b, D04, D05) are for LES simulations. The SADLES model is enabled in D05, with wind turbine locations marked by green and cyan markers. The FINO1 meteorological mast station is indicated by the red star. Refer to Table 2 for detailed domain dimensions.

The simulation period spans from 00:00 to 24:00 UTC on August 12, 2015. During this time frame, the first three domains (D01, D02, D03) run from 00:00 to establish mesoscale conditions, while the 200-meter WRF-LES domain (D04) begins at 12:00 UTC to serve as an intermediary between meso- and micro-scales. Finally, the 40-meter WRF-SADLES domain starts at 18:00 UTC to simulate turbine wakes. This period was selected because, towards its end, the wind direction shifts eastward, allowing for observation of wake effects using measurement data from the Fino1 station. Additionally, a low-level jet (LLJ), characterized by maximum wind speeds within the atmospheric boundary layer (ABL), is also observed during this timeframe.

Figure 10 compares wind speed and wind direction at 90 meters (hub height of the 5-MW Alpha Ventus wind turbines) from WRF simulations (D03: WRF-meso, D04: WRF-LES, and D05: WRF-SADLES) with observations from the FINO1 mast. Observations from both the mast data and LiDAR data suggest the wake effect from nearby turbines partially reduces wind speeds around 9:00, 13:00, 17:00, and 19:00 UTC, with reductions of approximately 2 m/s. From 22:00 onwards, the wakes exhibit a full effect, particularly when the wind direction approaches easterly (90 degrees), with wind speeds reduced from 6 to 8 m/s. While both data sources show good agreement in wind speed, there is a consistent difference of around 10 degrees in wind direction between the mast data and LiDAR data.

After a few hours of spin-up time, WRF-meso wind speeds agree well with observations when wakes do not affect the location. However, the intermediate LES domain (D05) underestimates wind speeds by about 2 m/s. Without the inclusion of the wind turbine model, neither WRF-meso nor WRF-LES captures the wake effect at the FINO1 location. In contrast, WRF-SADLES, which includes a wind turbine model, successfully simulates the full wake effect at FINO1. Wind speeds are

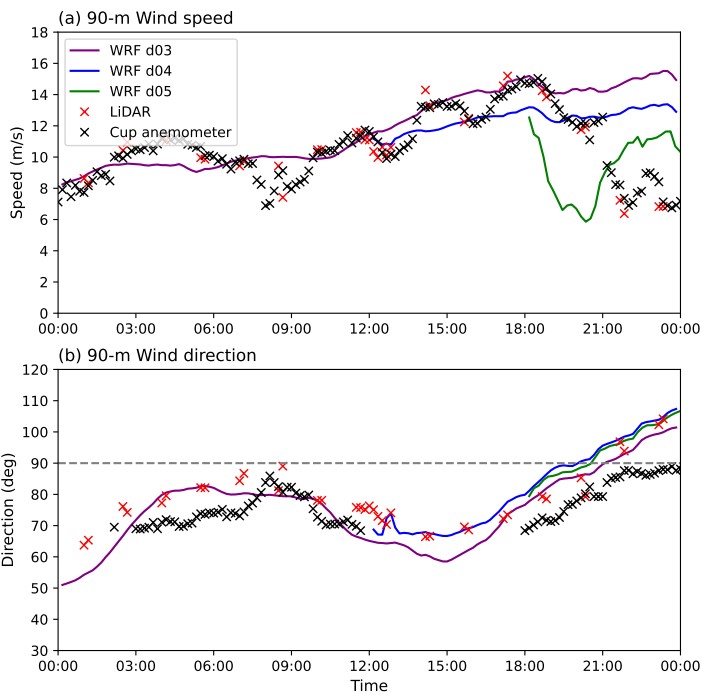

**Figure 10.** Time series of hub-height wind speed (a) and wind direction (b) at the FINO1 met-mast station on August 13, 2015, from 00:00 to 24:00 UTC, alongside data from WRF-meso (d01), WRF-LES (d02), WRF-SADLES (d03), LiDAR, and cup anemometers (Mast).

reduced to around 6 m/s, aligning well with observations. However, the timing of the wake impact differs, occurring between 19:00 and 21:00 UTC, roughly 2 hours earlier than observed.

To gain a clearer understanding of the discrepancies between WRF-SADLES simulations and observations, we examine the vertical profiles of wind speed (Fig. 11a) and wind direction (Fig. 11b) at two key points in time. The first is 20:30 UTC when the WRF-SADLES simulation shows the full wake effect. The second is 22:00 UTC when the full wake effect is observed in the data. At both times, the LiDAR data reveals a Low-Level Jet (LLJ) structure with a wind speed maximum of 17-20 m/s at around 300 meters. The observations of wind direction indicate wind veering, where the wind direction consistently turns clockwise with increasing height. However, within the overlapping region from 90 to 150 meters, some discrepancies exist between the two datasets. While the mast data exhibits consistency with the LiDAR data above 150 meters, the LiDAR wind directions below 150 meters may exhibit errors due to potential limitations in the retrieval algorithm or interference from the mast itself.

The mesoscale domain (D03) of the WRF model accurately replicates the observed LLJ) event, capturing both its magnitude and the height of the wind speed maximum. However, the WRF-LES and WRF-SADLES simulations produce a weaker

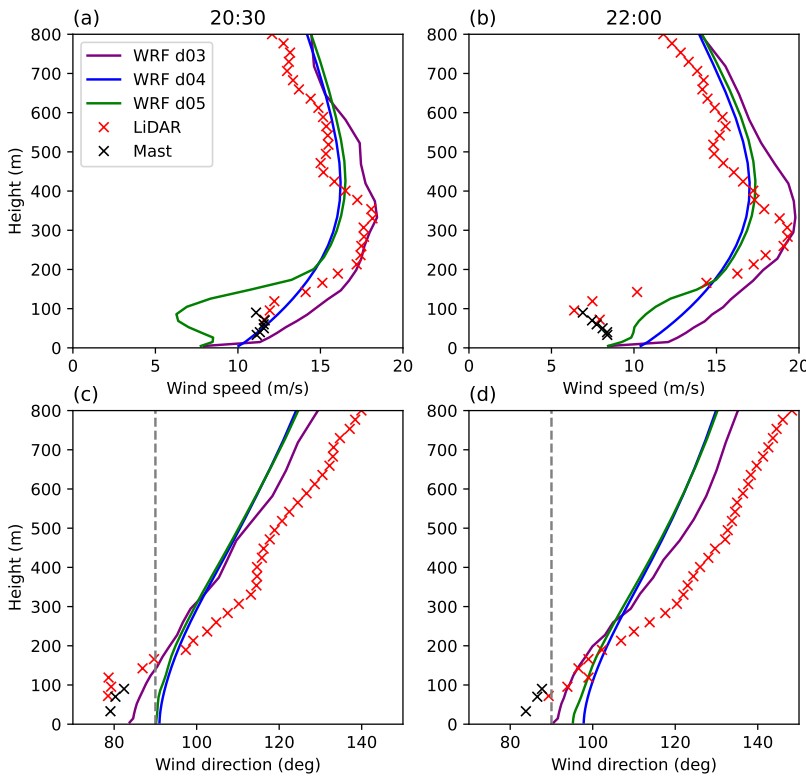

**Figure 11.** Vertical profiles of wind speed (a, b) and wind direction (c, d) recorded at the FINO1 met-mast station on August 13, 2015, at 20:30 UTC (a, c) and 22:00 UTC (b, d). The comparison includes data from WRF-meso (d01), WRF-LES (d02), WRF-SADLES (d03), LiDAR, and cup anemometers (Mast).

390    LLJ, with wind speeds 2-3 m/s lower. In terms of wind direction, the observations exhibit the highest vertical wind veer (approximately 8 degrees per 100 meters), followed by the mesoscale simulation (around 6 degrees per 100 meters). The WRF-LES simulations show the weakest vertical wind veer (about 4 degrees per 100 meters). The WRF-SADLES domain D05 performs slightly better than the WRF-LES domain D04 in capturing both wind speed and direction.

     The full wake effect is observed when the wind direction approaches 90 degrees (easterly). At such times, the wake from
395   the nearest Alpha Ventus turbine to the east reaches the FINO1 mast station. This is evident in both the observational wind speed (combining LiDAR and mast data) and the WRF-SADLES simulation, which show dips in wind speed. Notably, the wind speed profiles from the simulation and observations exhibit visually similar structures.

     Figure 12(b) shows a horizontal LiDAR scan from the FINO1 station at 23:00 UTC., at which the wind is blowing from the east and the full wake effect of a wind turbine from the Alpha Ventus is recorded. The LiDAR scan covers a circular

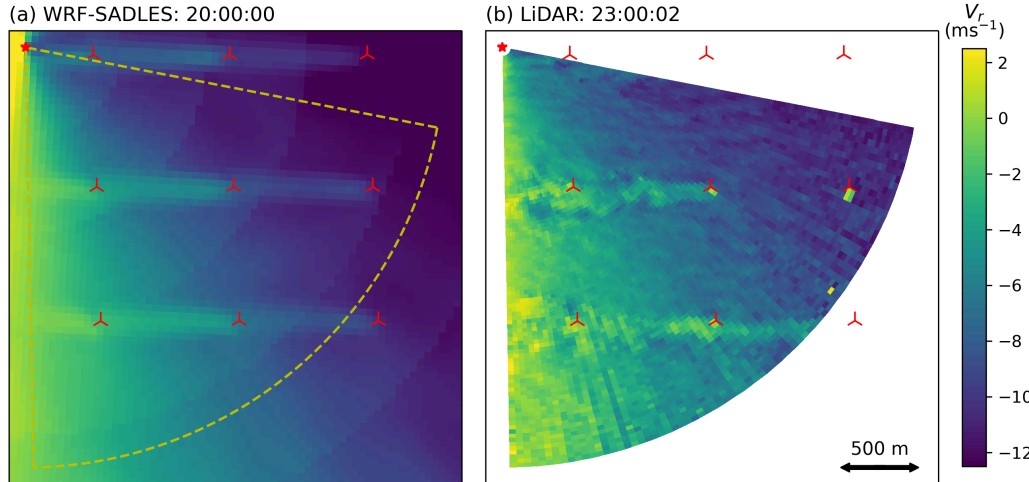

**Figure 12.** Radial wind speed from WRF-SADLES at 20:00 UTC (a) and horizontal LiDAR scans at the FINO1 station around 23:00 UTC (b).

sector extending from a height of 23.5 meters with a small elevation angle of 1.55 degrees, reaching a radius of 2500 meters and encompassing five turbines. The signal reaches a height of approximately 90 meters at the outer radius, corresponding to the hub height of the turbines. Turbine wakes are clearly visible, except for the middle turbine on the right, which is likely non-operational.

For comparison, in Fig. 12a, the simulated radial wind speed from the WRF-SADLES model at 23:00 UTC with an easterly wind direction is presented. Generally, there is good agreement between the model and LiDAR observations. Nevertheless, the WRF-SADLES wind speed distribution appears smoother due to its limited resolution compared to the LiDAR data

### 4.3 A brief example of farm-to-farm interaction

In this section, we briefly demonstrate the use of WRF-SADLES to simulate an example of farm-to-farm interaction. The LES simulations were conducted for domains D04 and D05 from 06:00 UTC to 12:00 UTC on September 24, 2016. The mesoscale domains commenced earlier at 00:00 UTC to initialize the environmental conditions. This timeframe was selected due to the relatively steady wind direction from the south-southwest, allowing farm-to-farm interactions between Alpha Ventus and the wind farm to the southwest. To quantify the influence of this farm, we set up two experiments: the first experiment (EXP1) incorporated all four wind farms, while the second experiment (EXP2) excluded all wind farms except Alpha Ventus.

Figure 13a and b shows an example snapshot of the wind speed at the Alpha Ventus hub height (90 m) for the 40-meter LES domain for the two experiments at 10:00 UTC on September 24, 2016. Thanks to the cell perturbation at the southern boundary, the turbulence quickly becomes fully developed after about two kilometers (or roughly ten percent of the domain

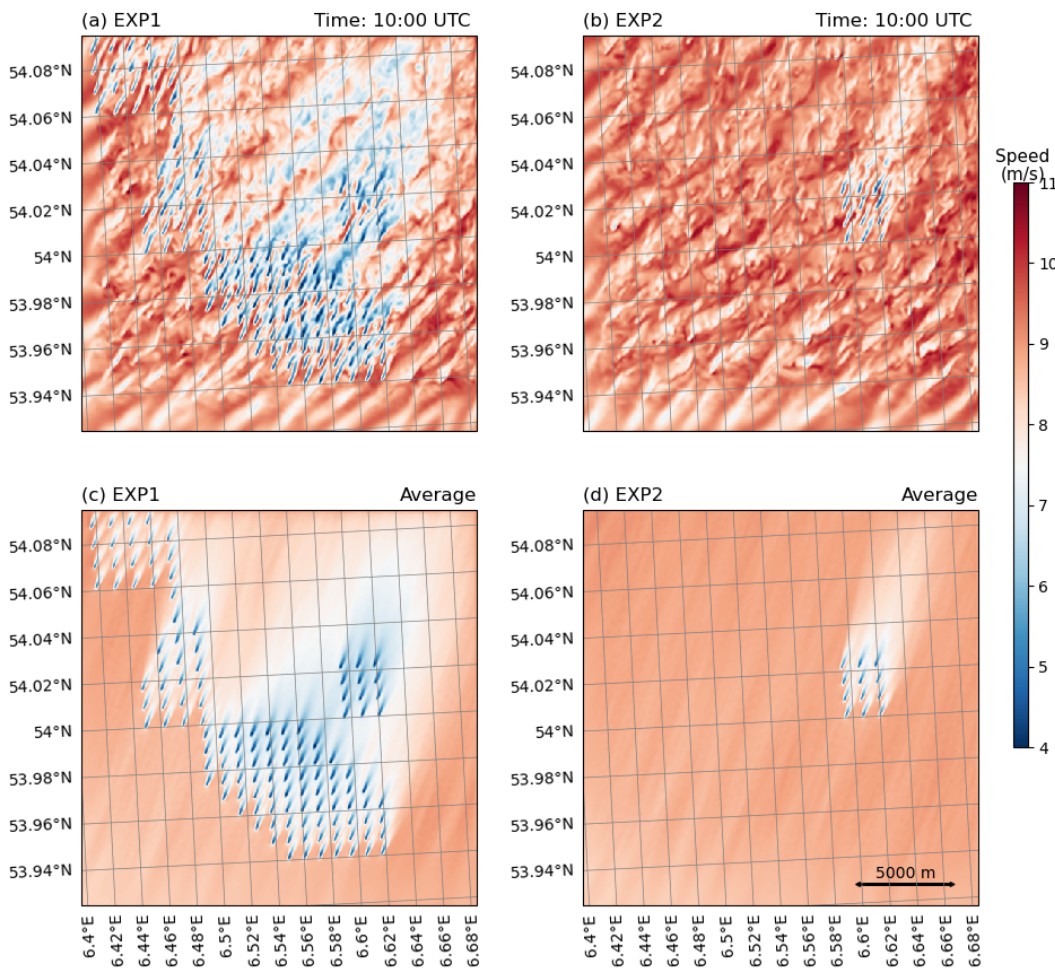

**Figure 13.** (a, b) A snapshot of wind speed in the 40-meter domain (D05) at the height of 90 meters at 10:00 UTC on September 24, 2016; (c, d) The 4-hour average wind speed (from 08:00 UTC to 12Z on September 24, 2016).

width) from the southern border. This allows the turbulence flow to become quasi-steady when it reaches the turbines in the wind farms. Such turbulence development is important because it affects the wake recovery behind the turbines.

Figure 13c and d shows the four-hour average wind speed from 08:00 to 12:00 UTC on September 24, 2016. During this time window, the wind direction is relatively steady from the South-Southwest, enabling us to address the potential impact of turbine wakes in the Alpha Ventus wind farm. Before coming to the wind farm, the average speed is slightly above 8 m/s and is distributed uniformly because of the small size of the domain. In EXP1, the south-southwesterly wind direction and the wake from a nearby wind farm significantly reduce the wind speed reaching Alpha Ventus (Fig. 13c). Conversely, with no upstream wind farm in EXP2, the averaged ambient wind speed remains nearly unchanged when approaching the Alpha Ventus wind farm. Consequently, the wake effect within Alpha Ventus is weaker in EXP1 compared to EXP2. This highlights the significant impact of farm-to-farm interaction, as evidenced by the larger difference in wind speed between the two scenarios compared to the variation within Alpha Ventus itself. In both experiments, the collective wave effects from the wind farm reduce the average wind speed to a distance over ten kilometers downstream.

Both experiments also show evidence of some intra-farm interaction where the wind speed deficits for turbines at the northeast corner are slightly smaller than those at the front south and west sides. However, due to the specific wind direction, the wakes generated by the turbines do not directly impact the turbines in the following rows. Consequently, the variation in wind speed deficit between turbines within the Alpha Ventus wind farm is minimal compared to the overall difference between the two experiments.

Figure 14 also shows the intra-farm interaction is small compared to farm-tom-farm interaction. For each experiment, the variation of the ambient wind speed is smaller than the difference between EXP1 and EXP2. The average ambient wind speed at the Alpha Ventus wind farm is significantly lower when the wind farm to the south presents (EXP1) compared to when there are none (EXP2). For EXP1, the average ambient wind speed is 7 m/s, as shown in Fig. 14a. In contrast, when there are no nearby wind farms (EXP2), the average ambient wind speed is 8.3 m/s, as shown in Fig. 14b. This represents a reduction in wind speed of about 16%. However, due to the non-linear nature of the turbine power curve, the power reduction resulting from the farm-to-farm interaction (Fig. 14) is larger. The average power for EXP2 is approximately 2.25 MW, while the average power for EXP1 is 1.4 MW, corresponding to a reduction of about 38%.

## 5   Discussion

The turbulence in the wake behind a wind turbine primarily arises from shear due to reduced wind speed in the wake region (Crespo et al., 1996; Quarton and Ainslie, 1990). In mesoscale modeling, incorporating turbulence kinetic energy (TKE) into wind farm parameterization is necessary due to the model's inability to resolve wakes at small scales (Fitch et al., 2012). Fitch et al. (2012) assumes that a portion of the extracted energy from the mean wind becomes power (related to thrust coefficient $C_T$), while the remainder becomes TKE, proportional to $C_T - C_P$. Given WRF-SADLES's target resolution of a few dozen meters, where the wake is partially resolved, some added subgrid-scale TKE may be necessary. However, testing Fitch's method in WRF-SADLES with subgrid-scale TKE using a TKE factor ($C_{TKE} = C_T - C_P$) revealed minimal influence on the far wake

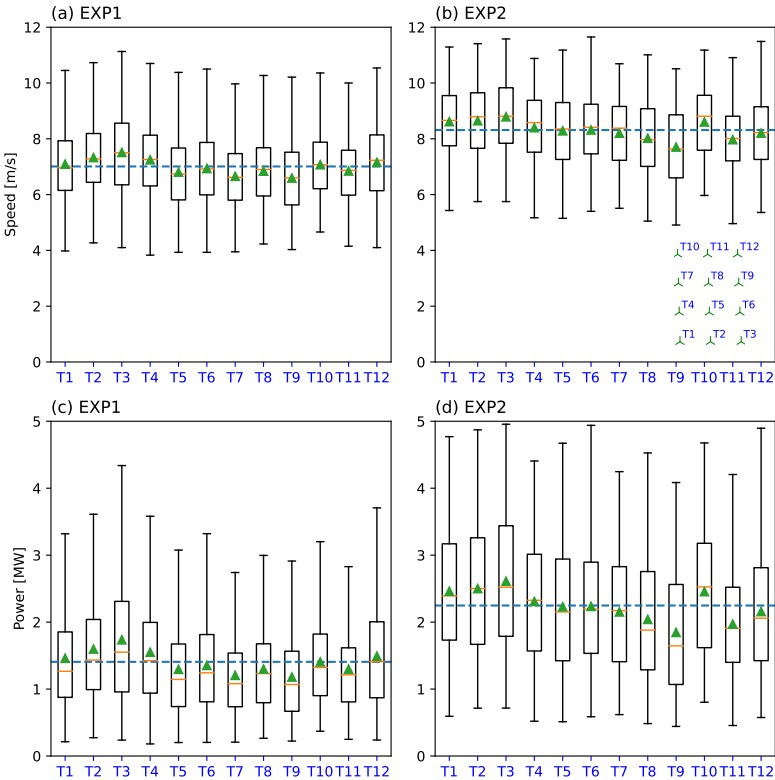

**Figure 14.** Box plots of 4 hours (from 08:00 to 12:00 UTC on September 24, 2016) for ambient wind speeds (a and b) and turbine powers (c and d) for the turbines in the Alpha Ventus wind farm. The average values (blue dashed lines) are 7 m/s and 8.3 m/s for (a) and (b), 1.4 MW and 2.25 MW for (c) and (d) respectively. The layout of the turbines in the Alpha Ventus wind farm is shown on the bottom right of panel (b).

structure, except near the turbine. This raises questions about the validity of Fitch's assumption, as the TKE downstream should relate to wake deficit, which is linked to $C_T$, not $C_T - C_P$. In an idealized scenario where all TKE extracted from the mean wind converts to power ($C_T = C_P$), according to Fitch, no TKE is generated, which is not appropriate. Thus, we imply that wind farm parameterization in mesoscale models should consider the relationship between added turbulence and $C_T$, alongside factors like stability conditions.

The mesoscale-to-microscale downscaling method proposed by Muñoz et al. Muñoz-Esparza et al. (2014) and in (Bakhoday-Paskyabi et al., 2022a) aimed to seamlessly transition atmospheric processes across scales. WRF-SADLES achieves this through nested simulations. However, in our case study (Fig. 11), WRF-LES simulations underestimated the strength of the low-level jet, with the vertical wind veer only half of the observed values. This underestimation occurred in both the WRF-LES domain (D04, 200-meter resolution) and the WRF-SADLES domain (D04, 40-meter resolution).

We attribute this limitation to the resolution used in domain D04, which falls within the gray zone or 'terra incognita' for numerical models (Wyngaard, 2004). This zone represents a transition region between mesoscale and microscale, where traditional boundary layer parameterizations and LES are not accurately applicable. Here, turbulence activities may be misrepresented, leading to simulation inaccuracies. While WRF-SADLES simulations show some improvement, the small domain size likely restricts its ability to fully correct the erroneous environmental conditions inherited from the inflow boundaries of

D04. We attempted to bypass this issue by skipping the 200-meter resolution domain. However, this led to WRF model crashes, possibly due to the large grid ratio (1:25) not being supported.

## 6   Conclusion

In this paper, we present our implementation of a Simple Actuator Disc model for Large Eddy Simulation (SADLES) within the Weather Research and Forecast model (WRF-SADLES). Unlike other previous implementations of wind turbine param-

eterization within WRF, such as the General Actuator Disc (GAD) (Mirocha et al., 2014; Kale et al., 2022), WRF-SADLES only requires the power curve and thrust coefficient curve—the same information used in wind farm parameterization (WFP Fitch et al., 2012) that is already included in WRF. The purpose of WRF-SADLES is to explicitly simulate the wakes of multiple wind farms in online nested downscaling applications from realistic atmospheric conditions. The target resolution of the WRF-SADLES is intermediate between the GAD model and the WFP, in which one turbine is represented by a few grid points.

WRF-SADLES employs the traditional actuator disc model, representing the turbine as a thin disc that generates thrust, slowing down ambient wind speed. In our idealized experiments with a single 5-MW turbine (Section 3), WRF-SADLES demonstrated good agreement compared to a dedicated LES model (the PALM model), which incorporates rotation in its actuator disc model, providing a more comprehensive representation. Interestingly, at our target resolution of 30 meters, WRF-SADLES exhibited better agreement with the 10-meter resolution than the PALM model.

We assessed two methods for evaluating the axial induction factor: direct evaluation (Option 1) using data points at and in front of the turbine, and inferred evaluation (Option 2) using 1D momentum theory. The results demonstrated strong agreement between the two methods.We recommend employing Option 2 due to its independence from model resolution and its ability to avoid potential issues associated with Option 1, as discussed in Section 2.1.

    Additionally, we experimented with adding subgrid-scale turbulence kinetic energy (TKE) at the actuator disc, similar to

the approach by Fitch et al. (2012). However, the effect of added TKE only influenced a short distance from the turbine, and the far wake structures, including wake deficit and turbulent intensity, remained similar regardless of the presence of added TKE. Therefore, we recommend deactivating this option (i.e. $f_{TKE} = 0$) for practical application, partly because the rationale behind the method does not reflect reality.

    We conducted a brief validation of WRF-SADLES using observational data from the FINO1 offshore meteorological mast

station. This data included measurements from a cup anemometer at 90 meters height, as well as vertical and horizontal wind profiles obtained by LiDAR. The simulation downscales the ERA5 global reanalysis from a coarse resolution (approximately 31 kilometers) to a turbine scale resolution (40 meters). This downscaling is achieved through a system of five nested domains,

with the outer three domains simulating mesoscale processes and the two inner domains simulating eddy turbulence. The results demonstrate good agreement in the wake deficit observed at the FINO1 location between WRF-SADLES and the actual observations. However, there is a discrepancy in the timing of the wake deficit occurrence. We attribute this error not to the turbine model itself, but to the intermediate 200-meter resolution LES domain. This domain serves as the transitional zone between mesoscale and microscale, where turbulence activity is not accurately represented.

Finally, we present an example of farm-to-farm interaction at the Alpha Ventus wind farm near the FINO1 station. Here, the wind farm to the southeast of Alpha Ventus leads to a reduction in ambient wind speed by approximately 16%, resulting in an average turbine power decrease of 38% during a 4-hour analysis window.

We've made our code openly available to promote further research in wind energy. Our code distribution includes our implementation of the cell perturbation method (Muñoz-Esparza et al., 2014), crucial for turbulence development in nested LES. While WRF-SADLES demonstrates promising capabilities for meso-to-micro downscaling, addressing issues in the transition resolutions will be crucial for enhancing wake predictions. Future development areas for WRF-SADLES could involve implementing yaw misalignment to enable wake deflection and investigating turbine yaw control strategies.

# Appendix A: Additional WRF namelist options

**Table A1.** Summary of WRF-SADLES namelist options

| Namelist options (`&physics`) | Default value | Description |
| --- | --- | --- |
| `sadles_opt` (max_domains) | 0 | =0 turn off SADLES for the current domain; =1 or 2: use Option 1 (direct) or Option 2 (inferred) for the induction factor, respectively |
| `sadles_startmin` (max_domains) | 0 | Time to start SADLES in minutes |
| `sadles_maxradius` * | 120 | Max turbine radius in meter, |
| `sadles_mindx` * | 20 | Min $\Delta x$ in meter, |
| `sadles_mindz` * | 20 | Min $\Delta z$ in meter, |
| `sadles_tkefact` | 0.5 | $f_{KTE}$, from 0 to 1 (see the text for the description) |
| `ideal_f` | 0.0001 | Coriolis force (`em_les` only), |
| `ideal_znt` | -1. | Surface roughness length (`em_les` only, only effective for positive values). |

\* These values are used to allocate arrays within the SADLES module.

**Table A2.** Summary of cell perturbation namelist options

| Namelist options (&cpert) | Default value | Description |
|---|---|---|
| cell_pert_xs (max_domains) | 0 | =1 will apply cell perturbation for the western boundary |
| cell_pert_xe (max_domains) | 0 | =1 will apply cell perturbation for the eastern boundary |
| cell_pert_ys (max_domains) | 0 | =1 will apply cell perturbation for the southern boundary |
| cell_pert_ye (max_domains) | 0 | =1 will apply cell perturbation for the northern boundary |
| cell_pert_magnitude | 0.5 | Magnitude of the cell perturbation |
| cell_pert_interval (max_domains) * | 320 | Interval to apply the perturbation in seconds |
| cell_pert_k1 | 8 | Bottom level of the transition layer for cell perturbation |
| cell_pert_k2 | 16 | Top level of the transition layer for cell perturbation |

* The interval should be approximately $\frac{8\Delta x}{U}$, where $U$ is the average wind speed during the simulation. For example, with $\Delta x = 200$ m and $U = 5$ m/s, the interval will be approximately $\frac{8(200 \text{ m})}{5 \text{ m/s}} = 320$ s.

*Code and data availability.*

Our WRF-SADLES initial release code with an example of idealized settings for the 30-meter simulation can be downloaded from: https://doi.org/10.5281/zenodo.10803669 (Bui, 2023). A short WRF-SADLES user's guide can be obtained from
WRF-SADLES's GitHub repository: https://github.com/haibuihoang/WRF-SADLES. The WRF-ARW model can be downloaded from https://github.com/wrf-model/WRF. The PALM model can be downloaded from https://gitlab.palm-model.org/releases/palm_model_system. The ERA5 hourly reanalysis can be downloaded from: https://cds.climate.copernicus.eu/. The information of turbines can be downloaded from: https://doi.org/10.5281/zenodo.4668613 (Larsén and Fischereit, 2021).

*Author contributions.* Hai Bui proposed and implemented the WRF-SADLES code, conducted the WRF-SADLES simulations, and wrote
the manuscript. Mostsafa provided turbine information, revised the manuscript, and assisted with several technical discussions. Mohammadreza contributed by writing the description of the PALM model, performing the PALM simulation, and assisting with manuscript revisions.

*Competing interests.*

All authors declare that they have no competing of interest.

*Acknowledgements.* The work is a part of the HIghly advanced Probabilistic design and Enhanced Reliability methods for high-value, cost-efficient offshore WIND (HIPERWIND) project, which has received funding from the European Union's Horizon 2020 Research and Innovation Programme under Grant Agreement No. 101006689. The simulations were performed on resources provided by project NN9871K by UNINETT Sigma2 - the National Infrastructure for High Performance Computing and Data Storage in Norway.

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
