# Peer review of "Implementation of a Simple Actuator Disc for Large Eddy Simulation in the Weather Research and Forecasting Model (WRF-SADLES-V1.2) for Wind Turbine Wake Simulation"

_EGUsphere, 2023_

## Referee Comment (RC1)

**1st review of ''Hai Bui et al. Implementation of a Simple Actuator Disc for Large Eddy Simulation (SADLES-V1.0) in the Weather Research and Forecasting Model (V4.3.1) for Wind Turbine Wake Simulation. Submitted to Egusphere, 2023''**

Summary

The work focuses on downscaling simulations for wind energy applications, and presents an implementation of the standard Actuator Disk Model (ADM-std) and the perturbation method (PM) in WRF-LES. In this context, two cases are presented. First, an idealized single turbine case is presented, where some options are tested. These simulations are compared with PALM simulations, serving as a pseudo-validation. Secondly a downscaling of the ERA5 reanalysis data around the Alpha Ventus wind farm is presented.

General comments

Overall, the research presented in the paper is very relevant and interesting to the field. However, the paper has major issues, which are unacceptable in a scientific publication manuscript:

- The manuscript has two repeated copy-pasted portions, one from line 59-65/66-72 and another from 256-259/260-264.
- Multiples figures have major issues. In Figure 1b the axes units are marked as kilometres when it should be meters. Also, the grid shown in Figure 1a makes no sense. Figure 3,5,9,10 don't have proper legends on the colormap.

In addition, a few other very important points must be addressed:

- The actuator model implemented is not new in itself and the novelty is only in its implementation in WRF-LES. The text should be very clear with this. This is often not clear e.g. lines 2-3. Also, each time SADLES is used in the text, it should be replaced by WRF-SADLES, e.g. line 7, or eventually WRF-LES-SAD, which seems more appropriate.
- In the community, BEM stand for Blade Element Momentum theory, which is not the same as the Blade Element theory alone which is used in AD+R and ALM, the text should be corrected to avoid any confusion, e.g. line 23,67, etc.
- The idea expressed in lines 113-118, that the rotation affects the wake recovery which justify the use of an additional subgrid-scale turbulence term since the rotation is not explicitly included, is not correct. Multiple studies have shown that the rotation is not important in LES with Actuator Models. The wake recovery mainly depends on the interaction between the wake with the incoming flow turbulence, including the turbulence generated by the wake shear itself, which means of course on how the simulation capture this accurately, in which the numerics, the sgs model, etc. play a critical role. Adding an additional artificial subgrid-scale turbulence term is fine, but it should be presented as so. Please mention the $f_{TKE}$ used.
- The strong use of instantaneous flow to analyse the flow is inappropriate. The instantaneous flow can be used for illustration but not for stong flow analyses and conclusions. For example, the statement lines 226-227 make no sense. Also, the statement lines 311-314 has no base/proof. In the same time, turbulence intensity plots are missing. Figures 5,6 and 10 must be reproduced with turbulence intensity.

These points should be addressed before continuing the review process.

---

## Referee Comment (RC2)

Review for GMD manuscript egusphere-2023-491 titled "Implementation of a Simple Actuator Disc for Large Eddy Simulation (SADLES-V1.0) in the Weather Research and Forecasting Model (V4.3.1) for Wind Turbine Wake Simulation" by Hai Bui et al.

**Summary:**

This manuscript presents a new wind turbine parameterization model called SADLES for WRF model, which strikes a balance between the accuracy of the GAD model and the computational efficiency of the WFP models. SADLES only requires power and thrust curves, which are already available in WRF. They validate the effectiveness of SADLES with PALM and also demonstrate a more realistic application by downscaling reanalysis data to investigate turbine-to-turbine and farm-to-farm interactions.

**Comments:**

The authors used SADLES and WRF-SADLES interchangeably in the manuscript. They should settle on using only one of the two names.

It is not clear on which option should users chose for the direct / inferred evaluation estimations of axial induction factor (a). Option 1 is used for the realistic application in the manuscript. Option 2, however, is more appropriate where the resolution is at a few hundred meters. It'll be great to have some recommendations to use each Options.

Figure 2 caption: "Power, thrust coefficient …" should read "Thrust coefficient …"

There are 2 red stars, one is FINO1, the other is not defined on Figure 8a.

---

## Referee Comment (RC3)

**Review Report: "Implementation of a Simple Actuator Disc for Large Eddy Simulation (SADLES-V1.0) in the Weather Research and Forecasting Model (V4.3.1) for Wind Turbine Wake Simulation"**

January 15, 2024

**Summary**

The paper titled "Implementation of a Simple Actuator Disc for Large Eddy Simulation (SADLES-V1.0) in the Weather Research and Forecasting Model (V4.3.1) for Wind Turbine Wake Simulation" introduces SADLES as a wind turbine model in WRF. The study aims for realistic downscaling of large eddy simulation and focuses on wind farm assessment. The major concerns revolve around the perceived lack of novelty, absence of radiation considerations, brevity in the discussion, and reliance on instantaneous flow analysis. Minor issues include duplicated text portions and unclear figure captions.

**Major Issues**

- **Novelty Clarification.** The paper should provide a clearer description of its novelty, particularly in the implementation of the actuator model in WRF-LES. Highlighting the unique aspects of this implementation would strengthen the paper's contribution.

- **Radiation Consideration.** The absence of radiation in the study, a crucial component in models like PALM, should be discussed as a limitation. It's recommended to reference relevant radiation-related papers (`https://doi.org/10.5194/gmd-15-145-2022` and `https://doi.org/10.5194/gmd-14-3095-2021`) to support this point.

- **Discussion Depth.** The discussion section is noted to be brief and superficial. Expanding this section to delve deeper into the implications and significance of the results would enhance the overall quality of the paper.

- **Instantaneous Flow Analysis.** Authors are advised not to rely solely on instantaneous flow-related analysis, emphasizing that in LES, such instantaneous flows may lack meaningful interpretation. Discussing the limitations and considerations regarding the choice of analyses would strengthen the paper.

- **Choice of Comparative Model.** The paper compares results with PALM, a numerical model. Authors should discuss the rationale for this choice and why they did not consider comparing results to experimental data or field measurements to enhance scientific validity.

**Minor Issues**

- **Text Repetition.** Duplicate copy-pasted portions in the manuscript (lines 59-65/66-72 and 256-259/260-264) should be addressed to ensure the clarity and flow of the manuscript.

- **Figure Captions.** The clarity of figure captions should be improved to enhance reader understanding. Clear, concise, and informative captions are essential for effective communication.

---

## Referee Comment (RC4)

The authors introduce a Simple Actuator Disc Low-order wind turbine model (SADLES) for Large-Eddy Simulation and implement its parameterization in the WRF model. This primarily addresses the downscaling simulation issue of wind farms within weather systems, striking a balance between the required accuracy for wind farm simulations and computational performance. Through validation in idealized scenarios and comparison cases, as well as application in the real-world Alpha Ventus offshore wind farm in Germany, the authors demonstrate the model's advantages in downscaling.

Comments:

1. There are two highly similar content sections in the manuscript that need careful inspection. One from lines 59-65/66-72, and another from lines 256-259/260-264.

2. The authors assume the use of the inferred evaluation method when the direct evaluation result exceeds 0.5, but this assumption lacks specific clarification. It's better if the authors consider including an analysis explaining why the direct evaluation method calculates the axial induction factor 'a' greater than 0.5.

3. From the perspective of the paper, there doesn't seem to be any difference between SADLES and WRF-SADLES. The authors should decide to use one term to refer to the model consistently.

4. In Figure 8a, there are two red stars, one indicating FINO1, and the other is undefined.

5. The authors should provide a more detailed explanation of the code implementation section.

6. In Figure 9 and Figure 10, the second subplot should be labeled as (b).

7. Appendix A: "Additonal WRF namelist options," the word "Additonal" should be adjusted to "Additional."

---

## Author Response (AR1)

**Responses to reviewer #1**

The work focuses on downscaling simulations for wind energy applications, and presents an implementation of the standard Actuator Disk Model (ADM-std) and the perturbation method (PM) in WRF-LES. In this context, two cases are presented. First, an idealized single turbine case is presented, where some options are tested. These simulations are compared with PALM simulations, serving as a pseudo-validation. Secondly a downscaling of the ERA5 reanalysis data around the Alpha Ventus wind farm is presented.

Thank you for your insightful review! Your suggestions have greatly contributed to the substantial and thorough revision of our manuscript. In response to your recommendations, we have incorporated new analyses, including an assessment of turbulent intensity (TI), and introduced additional idealized experiments to explore the impact of the subgrid-scale TKE factor. Furthermore, we have included a brief comparison with observational data from the FINO1 station and introduced a new discussion section to address the validity of the added subgrid-scale TKE and the limitations associated with the meso-to-micro scale transition resolution. We are confident that these revisions have significantly enhanced the manuscript compared to its initial version and trust that they adequately address your concerns. Below, we provide detailed responses to each of your points.

1. The manuscript has two repeated copy-pasted portions, one from line 59-65/66-72 and another from 256-259/260-264.

We greatly appreciate you bringing these typographical errors to our attention, which arose from internal revision oversights. We have removed the inappropriate parts and revising the remaining content.

2. Multiples figures have major issues. In Figure 1b the axes units are marked as kilometres when it should be meters. Also, the grid shown in Figure 1a makes no sense. Figure 3,5,9,10 don't have proper legends on the colormap.

We carefully checked all the figures. For Figure 1a, we've noted in the caption that it's just for illustration (so the vertical grid size stays the same while the WRF grids stretch). We also added a wind turbine drawing for clarity.

3. The actuator model implemented is not new in itself and the novelty is only in its implementation in WRF-LES. The text should be very clear with this. This is often not clear e.g. lines 2-3. Also, each time SADLES is used in the text, it should be replaced by WRF-SADLES, e.g. line 7, or eventually WRF-LES-SAD, which seems more appropriate

Thank you for your suggestion; we have revised the text accordingly. For instance, in the abstract, we have changed from "...introduce a new wind turbine model, the Simple Actuator Disc for Large Eddy Simulation (SADLES)," to "...present our implementation of a Simple Actuator Disc model for Large Eddy Simulation (SADLES)". We now use 'WRF-SADLES' instead of 'SADLES', except in a few instances where it's important to differentiate between the WRF and SADLES models. Regarding the abbreviation, We kindly request to maintain the name 'WRF-SADLES' instead of 'WRF-LES-SAD' due to our preference.

> 4. In the community, BEM stand for Blade Element Momentum theory, which is not the same as the Blade Element theory alone which is used in AD+R and ALM, the text should be corrected to avoid any confusion, e.g. line 23,67, etc.

Thank you very much, we have check addressed in the text the avoid any confusion.

> 5. The idea expressed in lines 113-118, that the rotation affects the wake recovery which justify the use of an additional subgrid-scale turbulence term since the rotation is not explicitly included, is not correct. Multiple studies have shown that the rotation is not important in LES with Actuator Models. The wake recovery mainly depends on the interaction between the wake with the incoming flow turbulence, including the turbulence generated by the wake shear itself, which means of course on how the simulation capture this accurately, in which the numerics, the sgs model, etc. play a critical role. Adding an additional artificial subgrid-scale turbulence term is fine, but it should be presented as so. Please mention the fTKE used.

Thank you for your valuable suggestion. To tackle this issue, we conducted four new additional experiments to explore the impact of the added TKE. The results indicate that the inclusion of subgrid-scale TKE does not significantly influence the outcomes. This added term can be managed in WRF-SADLES through the 'sadles_tkefact' option, for which we have provided the recommended value (sadles_tkefact=0) in the manuscript. We also discussed the appropriateness of the term $(C\_t-C\_p)$ in the new Discussion section.

> 6. The strong use of instantaneous flow to analyse the flow is inappropriate. The instantaneous flow can be used for illustration but not for stong flow analyses and conclusions. For example, the statement lines 226-227 make no sense. Also, the statement lines 311-314 has no base/proof. In the same time, turbulence intensity plots are missing. Figures 5,6 and 10 must be reproduced with turbulence intensity.

We appreciate your suggestion! We have implemented changes accordingly. For the idealized cases, we removed the instantaneous plots and included analysis and discussion of turbulent intensity (TI). In the case of realistic data, we conducted a new case study, compared it with observations, and introduced three new figures. Regarding the content on farm-to-farm example, we retained the instantaneous snapshots for visual illustration and combined four

figures from the previous version into two. We aim to keep this section concise since there's no reference simulation for comparison, and it primarily serves as an illustrative example of WRF-SADLES application.

**Responses to reviewer #2**

> This manuscript presents a new wind turbine parameterization model called SADLES for WRF model, which strikes a balance between the accuracy of the GAD model and the computaonal efficiency of the WFP models. SADLES only requires power and thrust curves, which are already available in WRF. They validate the effectiveness of SADLES with PALM and also demonstrate a more realistic application by downscaling reanalysis data to investigate turbine-to-turbine and farm-to-farm interactions.

Thank you for your review! We've incorporated your suggestions and substantially improved our manuscript. This includes new experiments, analysis of turbulent intensity (TI), and a brief comparison with observational data. Additionally, we've added a new discussion section addressing the validity of the added subgrid-scale TKE and the meso-to-micro downscaling method. We believe these revisions greatly enhance the manuscript and adequately address your concerns. Below, we provide detailed responses to your points.

> The authors used SADLES and WRF-SADLES interchangeably in the manuscript. They should settle on using only one of the two names.

Thank you for your suggestion. We have edited the manuscript as suggested.

> It is not clear on which option should users chose for the direct / inferred evaluation estimations of axial inducion factor (a). Option 1 is used for the realistic application in the manuscript. Option 2, however, is more appropriate where the resolution is at a few hundred meters. It'll be great to have some recommendations to use each Options.

We have carried out additional experiments and provided the recommendation options in the text. Specificly, The recommended options is: Option 2 for axial induction factor (Line 478), and f_TKE=0 for the added subgrid-scale TKE (Line 483).

> Figure 2 caption: "Power, thrust coefficient …" should read "Thrust coefficient …" There are 2 red stars, one is FINO1, the other is not defined on Figure 8a.

We have clarified the text in Fig.2's caption. Regarding Fig. 8, we have replotted by removing the other star and revised the caption.

**Responses to reviewer #3**

> The paper titled "Implementation of a Simple Actuator Disc for Large Eddy Simulation (SADLES-V1.0) in the Weather Research and Forecasting Model (V4.3.1) for Wind Turbine Wake Simulation" introduces SADLES as a wind turbine model in WRF. The study aims for realistic downscaling of large eddy simulation and focuses on wind farm assessment. The major concerns revolve around the perceived lack of novelty, absence of radiation considerations, brevity in the discussion, and reliance on instantaneous flow analysis. Minor issues include duplicated text portions and unclear figure captions.

Thank you for your review! We've integrated your suggestions, resulting in significant enhancements to our manuscript. This encompasses the addition of new experiments, analysis of turbulent intensity (TI), and a concise comparison with observational data. Furthermore, we've introduced a new discussion section that delves into the validity of the added subgrid-scale TKE and the meso-to-micro downscaling method. Alongside this, we've included a newly structured Discussion section. We are confident that these revisions elevate the quality of the manuscript and effectively address your concerns. Please find detailed responses to your points below.

> 1. Novelty Clarication. The paper should provide a clearer description of its novelty, particularly in the implementation of the actuator model in WRF-LES. Highlighting the unique aspects of this implementation would strengthen the paper's contribution.

Thank you for your suggestions. While WRF-SADLES is rooted in the traditional actuator disc model, its novelty lies in its integration within the widely utilized WRF model in atmospheric science. To clarify this distinction, we have adjusted the language in the abstract and conclusion, shifting from "...introduce a new wind turbine model, the Simple Actuator Disc for Large Eddy Simulation (SADLES)," to "...present our implementation of a Simple Actuator Disc model for Large Eddy Simulation (SADLES)." This modification aims to prevent any potential confusion.

> 2. Radiation Consideration. The absence of radiation in the study, a crucial component in models like PALM, should be discussed as a limitation. It's recommended to reference relevant radiation-related papers (https://doi.org/10.5194/gmd-15-145-2022 and https://doi.org/10.5194/gmd-14-3095-2021) to support this point.

Thank you for your comments. Radiation is indeed a critical component in weather numerical models, alongside factors such as cumulus and microphysics. In our realistic data, we employed the traditional scheme (RRTMG) as detailed in the text. However, for standard idealized LES simulations, we disabled radiation and other physical processes to simplify the setup and facilitate result interpretation. In fact, the impact of radiation in our idealized experiments is indirectly accounted for through the idealized surface turbulence heat flux. While the role of surface turbulence heat flux on turbulence and wake properties is an important and intriguing topic, it falls beyond the scope of our developmental paper.

> 3. Discussion Depth. The discussion section is noted to be brief and super cial. Expanding this section to delve deeper into the implications and signi cance of the results would enhance the overall quality of the paper.

Thank you for your valuable suggestion. We've carefully reviewed the paper and added a new Discussion section to explore aspects of WRF-SADLES, including the use of subgrid-scale added TKE and the transition resolution of the downscaling method.

> 4. Instantaneous Flow Analysis. Authors are advised not to rely solely on instan- taneous flow-related analysis, emphasizing that in LES, such instantaneous flows may lack meaningful interpretation. Discussing the limitations and considerations regarding the choice of analyses would strengthen the paper.

We appreciate your suggestion. We have removed the plot and discussion related to the idealized instantaneous flow. Instead, we added four new experiments focusing on the effect of the added subgrid-scale TKE and included analysis and discussion on turbulence intensity. For the realistic case, we retained one figure for illustrative purposes but revised the text to be more concise.

> 5. Choice of Comparative Model. The paper compares results with PALM, a numerical model. Authors should discuss the rationale for this choice and why they did not consider comparing results to experimental data or field measurements to enhance scientic validity.

Thank you for your suggestion. In addition to incorporating the discussion related to the reasons for using PALM (Lines 79-82), we introduced a new section (Section 4.2) comparing WRF-SADLES with observations, including cup anemometer and LiDAR data. We believe that these additions enhance the scientific validity of our manuscript substantially.

> 6. Text Repetition. Duplicate copy-pasted portions in the manuscript (lines 59-65/66-72 and 256-259/260-264) should be addressed to ensure the clarity and ow of the manuscript. Figure Captions. The clarity of figure captions should be improved to enhance reader understanding. Clear, concise, and informative captions are essential for effective communication.

Thank you very much! We have extensively revised the text, figures, and captions to create a more concise and clear manuscript.

**Responses to reviewer #4**

> The authors introduce a Simple Actuator Disc Low-order wind turbine model (SADLES) for Large Eddy Simulation and implement its parameterization in the WRF model. This primarily addresses the downscaling simulation issue of wind farms within weather systems, striking a balance between the required accuracy for wind farm simulations and computational performance. Through validation in idealized scenarios and comparison cases, as well as application in the real world Alpha Ventus offshore wind farm in Germany, the authors demonstrate the model's advantages in downscaling.

Thank you for your review! We have provided detailed responses to your points below. Additionally, we have thoroughly revised the manuscript, incorporating four new idealized experiments and one realistic case study with a comparison to observational data. Furthermore, we have conducted new analyses and added new plots. We believe these changes have significantly improved the manuscript.

> 1. There are two highly similar content sections in the manuscript that need careful inspection. One from lines 59 65/66 72, and another from lines 256 259/260 264..

Thank you for highlighting the typos identified during our internal revision. We have carefully reviewed the paper and addressed these issues.

> 2. The authors assume the use of the inferred evaluation method when the direct evaluation result exceeds 0.5, but this assumption lacks specific clarification. It's better if the authors consider including an analysis explaining why the direct evaluation method calculates the axial inductin factor 'a' greater than 0.5.

In the 1-D momentum method, the axial induction factor 'a' cannot exceed 0.5, as it suggests wind in the wake blowing against the mean wind. However, the model using Option can lead to this scenario, potentially causing the model to crash. We observed this issue during some of our test experiments. To address this, we have revised the relevant paragraph (Lines 134-136) to clarify: "By applying this formula, 'a' can exceed 0.5, indicating wind behind the turbine opposing the ambient wind. This is nonphysical and may result in model instability."

> 3. From the perspective of the paper, there doesn't seem to be any difference between SADLES and WRF-SADLES. The authors should decide to use one term to refer to the model consistently.

Thank you for your suggestion. We have revised the paper to clarify the issues. It's important to note that SADLES is the name of the module implemented in WRF, while WRF-SADLES refers to the entire package, which also includes the cell perturbation model. Throughout the paper, we have replaced most instances of 'SADLES' with 'WRF-SADLES', except in a few places where it is necessary to distinguish between the two terms

> 4. In Figure 8a, there are two red stars, one indicating FINO1, and the other is undefined.

We have replotted the figure and addressed the issue.

> 5. The authors should provide a more detailed explanation of the code implementation section.

Thank you for your suggestion, we added some more detailed about the code implmentation (Lines 155-167).

> 6. In Figure 9 and Figure 10, the second subplot should be labeled as (b).

We combined the two figures into one (which is now Figure 13) and addressed the issue.

> 7. Appendix A: "Additonal WRF namelist options," the word "Additonal" should be adjusted to "Additional"

Thank you a lot, we fixed the typo.

---

## Referee Report (RR1)

**2nd review of ''Hai Bui et al. Implementation of a Simple Actuator Disc for Large Eddy in the Weather Research and Forecasting Model (WRF-SADLES-V1.2) for Wind Turbine Wake Simulation.**

Summary

The authors implemented substantial changes in the manuscript. Overall, the manuscript has been improved. However, a few critical changes and a few other minor changes should be done before publication. Some of the points were already mentioned in the 1st review but not corrected properly.

Critical comments (general):

- In the AD+R and ALM the momentum theory is not used, because this part is taken care by the CFD simulation itself (corresponding mainly the effect of the turbine induction), and therefore it is wrong to say that they use the BEM theory (where the M stand for the momentum theory), they only use the BE theory. In the text, many wrong references to « BEM » are still present. This was already discussed in the 1st review. The text must be modified accordingly, e.g. lines 25,63,212,255. Also, « BE theory » seems a better formulation than « BMT » as proposed line 25.

- The idea that the simple AD (SAD) has weaker resolution requirements than the AD+R is not supported by the literature. In fact, the authors already agree with this, since they write « at least a few grid points across the rotor » for the AD+R (line 64), which is exactly the resolution targeted for the SAD in this paper (tens of meters). This argument can therefore not be used. The argument regarding the availability of the turbine data (line 65-66) is correct and sufficient. The text must be modified accordingly, e.g. lines 72,474.

- Multiple studies have shown that the rotation is not critical in LES with Actuator Models. This was already discussed in the 1st review. The critical difference between the SAD implemented here and the AD+R used in the PALM simulations is the force distribution on the rotor (homogeneous or heterogeneous respectively) and the way these forces are computed (using the thrust coefficient or the BE theory respectively), not the rotation. A SAD like model can have rotation included using the power coefficient, and we could imagine an Actuator Disk model based on the BE theory but without rotation (so similar to the AD+R of PALM but without rotation). The text must be modified accordingly, e.g. lines 114-115,246-247,477-478. Line 242, remove « (e.g. simple actuator disk versus actuator disk with rotation) », not needed and most likely not the most important point.

- The paragraph lines 111-116 is still very confusing. Referring to my 1st review: « The idea expressed in lines 113-118 [lines from V1], that the rotation affects the wake recovery which justify the use of an additional subgrid-scale turbulence term since the rotation is not explicitly included, is not correct. ». In fact, the authors already agree with this, in lines 443-454. In this context, lines 443-448 are well adapted and lines 111-116 should be adapted using lines 443-448.

- I would suggest moving lines 443-448 to lines 111-116, lines 449-454 at line 298, and lines 455-467 at line 397 (with the appropriate modification needed).

- In section 3.2, Lines 228-252 and 265-279, make strong statements which are difficult to justify based only on contours, e.g. « slower wake recovery » line 231. Profiles are better suited for these types of statements. To solve this issue, the profiles (Figs 5 and 7) could be presented at the same time as the contours (respectively Fig 4 and 6) and discussed together. Also, the profiles for the induction optimisation should be added (contours could be removed if needed, profiles are more important than contours for a precise discussion). Fig 3 and 4 could be merged and discussed all together.

Critical comments (specific):

- Lines 231-232: « indicating potentially stronger turbulence activities », in fact, it is the opposite, slower recovery and smaller rate of wake expansion are in general associated with lower background turbulence intensity. Must be corrected or reformulated.
- Line 27: « rotating circular disk », not appropriate, must be corrected.

- Line 145: « within the interval […] chosen boundaries », unclear, must be reformulated.
- Line 233-234. The idea behind the sentence is very strange formulated as such, change « which is used » with « for ».
- Lines 275-276: « shear production », not appropriate, Line 283 « shear production of turbulence », not appropriate. Must be reformulated.
- Lines 301-302: not clear how the spectra are computed, from what I understand, not sure it is appropriate for the targeted goal. Clarify.
- Lines 405-406, Fig 12: I am not convinced by the « limited resolution » argument (referring to spatial resolution I guess), the flow from the simulation looks like there is no turbulence, any comments on that?
- Lines 482-483, « independence from model resolution »: does not seem supported by the results and does not seem correct, must be changed/removed.
- Fig 1: the vertical resolution in Fig 1.a. still do not match the legend (dz=30m) and Fig 1.b. This was already discussed in the 1st review.
- Fig 10: remove « alongside», Fig 11: modify second sentence of caption, with « , data from […] » similar to caption of Fig 10.

Additional comments:

- Line 3: « downscaling of large eddy simulations », no so clear in the context.
- Line 20: « dynamic loading », no so clear in the context, not really needed.
- Line 36: « such as cyclones an fronts », no so relevant, not really needed.
- Lines 57-58: « (Fitch et al., 2012) only requires », replace with : « the method used by Fitch et al. ».
- Line 74: « the LES downscaling approach », formulation could be improved.
- Line 75: « necessary for generating turbulence », replace with something like « allows faster development of turbulence ».
- Lines 79-80: could be improved, e.g. with something like « an idealized case is used to compare the simulation results from WRF-SADLES with PALM ».
- Lines 89-90: does not look appropriate with Section 5. Section 6 is missing.
- Lines 97-98: « The wind speed at the rotor […] », introduce this at line 108, before actually using the wind speed at the rotor in the equation.
- L115: « we adopt », replace with « we test ».
- L 144: « improve the realism of turbulent representation », not appropriate.
- L172-173: « models, known for their realistic simulation », not clear (we could think the statement also includes WRF-SADLES), improve formulation, could be simply removed.
- L173-174: indicate also the resolution with respect to the rotor diameter », i.e. how many points per rotor diameter.
- Line 184: « an aspect ratio of 2:1 », add the directions (could be vertical at this moment in the text).
- Line 193: « zonal direction », not clear.
- Line 207: reference in bad format.
- Line 222: express time in hours like for the WRF simulation.
- Line 230: « suggesting a dependency of momentum fluxes on model solution », not sure to follow, may need more explanation or reformulation.
- Line 232: « conversely », not adapted.
- Line 241: « methodological », maybe « numerical »?
- Line 245-246: improve, e.g. with something like « PALM simulation exhibits stronger wake expansion than the WRF simulation ».
- Line 250: « but one possibility is related », not very elegant, could be improved.
- Line 252: « (2D) », remove parenthesis.
- Line 253: « agree around 50% », not adapted, remove.
- Line 280: « vertical structure », modify with « «vertical profile ».
- Line 290: « brings them closer », not elegant.
- Line 388: « LLJ) », remove parenthesis.
- Line 398: »UTC., », remove « . ».

- Line 416: « become fully developed after about two kms », justify the statement or make less precise statement.
- Line 452: « we imply », not appropriate.
- Line 465: « model crashes », not so elegant.
- In figures 4,6,7, « similar than in fig […] », copy-paste the appropriate description, each figure should be self descriptive.
- Fig 5, « evaluation point » maybe not so clear (not really a point since it's a profile), « location » could be better.
- Fig 12, « radial wind speed », not so clear to me in this context.
- Fig 14: the layout in panel in (b) could be incorporated in Fig 9, and the y axis of Fig (a) and (b) could be rescaled.
- Table 2: « real-data », not very elegant.

Remarks/questions:

- Why only potential temperature in the perturbation method and not the wind speed components?
- Why using a precursor simulation in PALM and no precursor in WRF simulation ?

---

## Author Response (AR3)

**Responses to referee #1**

The authors implemented substantial changes in the manuscript. Overall, the manuscript has been improved. However, a few critical changes and a few other minor changes should be done before publication. Some of the points were already mentioned in the 1st review but not corrected properly..

Thank you for your constructive review! In this revision, we carefully revised our manuscript to address your concerns. The bellows are out point-to-point response.

**Critical comments (general):**

In the AD+R and ALM the momentum theory is not used, because this part is taken care by the CFD simulation itself (corresponding mainly the effect of the turbine induction), and therefore it is wrong to say that they use the BEM theory (where the M stand for the momentum theory), they only use the BE theory. In the text, many wrong references to « BEM » are still present. This was already discussed in the 1st review. The text must be modified accordingly, e.g. lines 25,63,212,255. Also, « BE theory » seems a better formulation than « BMT » as proposed line 25.

Thank you for your comments. We follow your suggestion of using abbrevation "BE theory". Upon careful examination of the methods employed in WRF-GAD and PALM, we have noted that while the PALM model uses the BE theory, the WRF-GAD model uses the Blade Element Momentum (BEM) theory, which combines the BE theory with momentum theory through induction factors. For instance, in WRF-GAD paper (Mirocha et al. 2014), the authors explicitly state that "The GAD model implemented into WRF follows the generalization of the Blade Element Momentum theory of Glauert...". In essence, their model integrates stream tube dynamics (momentum theory) with blade aerodynamics (BE theory) (See their apprendix). Thus, we have revised the entire paragraph and other places on the paper to provide clarity on this matter.

Ref: Mirocha, J. D., Kosovic, B., Aitken, M. L., & Lundquist, J. K. (2014). Implementation of a generalized actuator disk wind turbine model into the weather research and forecasting model for large-eddy simulation applications. *Journal of Renewable and Sustainable Energy*, 6(1).

The idea that the simple AD (SAD) has weaker resolution requirements than the AD+R is not supported by the literature. In fact, the authors already agree with this, since they write « at least a few grid points across the rotor » for the AD+R (line 64), which is exactly the resolution targeted for the SAD in this paper (tens of meters). This argument can therefore not be used. The argument regarding the availability of the turbine data (line 65-66) is correct and sufficient. The text must be modified accordingly, e.g. lines 72,474.

> The idea that the simple AD (SAD) has weaker resolution requirements than the AD+R is not supported by the literature. In fact, the authors already agree with this, since they write « at least a few grid points across the rotor » for the AD+R (line 64), which is exactly the resolution targeted for the SAD in this paper (tens of meters). This argument can therefore not be used. The argument regarding the availability of the turbine data (line 65-66) is correct and sufficient. The text must be modified accordingly, e.g. lines 72,474.

Thank you for your comments. We removed the argument as you suggested and revised the relevant paragraphs. We put the notions about the resolution in these two sentences: "While many WRF-GAD applications typically employ fine resolutions of a few meters (e.g. Mirocha et al., 2014; Arthur et al., 2020; Kale et al., 2022), such high resolutions are computationally expensive for realistic downscaling problems involving large domains with multiple wind farms. Therefore, WRF-SADLES is also tested with coarser resolutions (specifically, 30 and 40 meters) to achieve a more practical balance between computational cost and wake resolution".

> Multiple studies have shown that the rotation is not critical in LES with Actuator Models. This was already discussed in the 1st review. The critical difference between the SAD implemented here and the AD+R used in the PALM simulations is the force distribution on the rotor (homogeneous or heterogeneous respectively) and the way these forces are computed (using the thrust coefficient or the BE theory respectively), not the rotation. A SAD like model can have rotation included using the power coefficient, and we could imagine an Actuator Disk model based on the BE theory but without rotation (so similar to the AD+R of PALM but without rotation). The text must be modified accordingly, e.g. lines 114-115,246-247,477-478. Line 242, remove « (e.g. simple actuator disk versus actuator disk with rotation) », not needed and most likely not the most important point.

> The paragraph lines 111-116 is still very confusing. Referring to my 1st review: « The idea expressed in lines 113-118 [lines from V1], that the rotation affects the wake recovery which justify the use of an additional subgrid-scale turbulence term since the rotation is not explicitly included, is not correct. ». In fact, the authors already agree with this, in lines 443-454. In this context, lines 443-448 are well adapted and lines 111-116 should be adapted using lines 443-448.

> I would suggest moving lines 443-448 to lines 111-116, lines 449-454 at line 298, and lines 455-467 at line 397 (with the appropriate modification needed).

We thank you for your clarification and comments. We carefully revised the paper related to your comment as follows:

- We moved the discussion about this from lines 443-448 to lines 111-116 and revised the text (Lines 116-120 in revised version).

- In Line 242, we removed « (e.g. simple actuator disk versus actuator disk with rotation) » as suggested.

- In lines 246-258, we change the text from "...the absence of the rotational effect in WRF-SADLES, which is included in PALM.", to "...how the thrust forces are calculated differently in the BE method (PALM) and the momentum theory (WRF-SADLES)."

- We moved lines 449-454 to line 298 and adapted the text as suggested (Lines 291-294 in revised version)

- We removed "which incorporates rotation in its actuator disc model, providing a more comprehensive representation" in lines 477-478, replaced by "whose actuator disc model uses the blade element theory, providing a more comprehensive representation".

- We moved Lines 455-467 to line 397 with modification to adapted to the text flow (Lines 376-387).

> In section 3.2, Lines 228-252 and 265-279, make strong statements which are difficult to justify based only on contours, e.g. « slower wake recovery » line 231. Profiles are better suited for these types of statements. To solve this issue, the profiles (Figs 5 and 7) could be presented at the same time as the contours (respectively Fig 4 and 6) and discussed together. Also, the profiles for the induction optimisation should be added (contours could be removed if needed, profiles are more important than contours for a precise discussion). Fig 3 and 4 could be merged and discussed all together.

Thank you for your suggestions! As suggested, we merged Fig. 3 and 4 together and added wind deficit profile to Fig. 5 (Fig. 4 in the revised version). We carefully chose the lines colors to make sure that they are color blind friendly. After that we revised the whole section as suggested (i.e. discuss the contour and profile figures together). We believe the text becomes more concise and comprehensive!

**Critical comments (specific):**

> Lines 231-232: « indicating potentially stronger turbulence activities », in fact, it is the opposite, slower recovery and smaller rate of wake expansion are in general associated with lower background turbulence intensity. Must be corrected or reformulated

In our original text, it is "indicating potentially stronger turbulence activities at lower resolutions.", which is not incorrect, but easily causes confusion. Nevetherles, this is removed in our revised text.

> Line 27: « rotating circular disk », not appropriate, must be corrected.

We replaced the phrase by "permeable disc"

> Line 145: « within the interval […] chosen boundaries », unclear, must be reformulated.

For clarity, we extend the sentence to "The method introduces a random perturbation of potential temperature within the interval of -0.5K to 0.5K to three cells near the inflow boundaries. Each cell is an 8x8 grid points square in the horizontal plane, and the same perturbation is applied to each cell. Therefore, the total perturbation zone extends 24 grid points from the inflow boundaries."

> Line 233-234. The idea behind the sentence is very strange formulated as such, change « which is used » with « for ».

We changed the text as suggested.

> Lines 275-276: « shear production », not appropriate, Line 283 « shear production of turbulence », not appropriate. Must be reformulated.

We changed the phrases to "increased wind shear at these boundaries" and "turbulence production associated with the wake shear" respectively.

> Lines 301-302: not clear how the spectra are computed, from what I understand, not sure it is appropriate for the targeted goal. Clarify.

We shorten these discussions and move them to the beginning of Section 3.2 to briefly compare the turbulence characteristic of PALM and WRF at the two resolutions, showing that the two models are comparable. We added more explaination to the caption of the Figre (Now Fig. 3).

> Lines 405-406, Fig 12: I am not convinced by the « limited resolution » argument (referring to spatial resolution I guess), the flow from the simulation looks like there is no turbulence, any comments on that?

Yes, we mean the spatial resolution. As showin the Fig. 3, coarser solution under resolve the turbulence for small eddies. For clarity, we revised the sentence as "This is likely attributable to the coarse resolution to capture the small-scale turbulence explicitly."

Note that the figure is zoomed in to compare with the LiDAR. Here is the snapshot of zonal wind

without clipping.

> Lines 482-483, « independence from model resolution »: does not seem supported by the results and does not seem correct, must be changed/removed.

We removed the argument as suggested.

> Fig 1: the vertical resolution in Fig 1.a. still do not match the legend (dz=30m) and Fig 1.b. This was already discussed in the 1st review.

Thank you for point it out again (we are a little bit confused in the 1s review)! Yes, there is a missmatch, dz should be 20m instead and we corrected in the caption.

> Fig 10: remove « alongside», Fig 11: modify second sentence of caption, with « , data from […] » similar to caption of Fig 10.

We editted the captions as suggested.

**Additional comments:**

> Line 3: « downscaling of large eddy simulations », not so clear in the context.

We chanced the sentence to "The WRF-SADLES model supports both idealized studies and realistic applications through downscaling from real data, with a focus on resolutions of tens of meters."

> Line 20: « dynamic loading », no so clear in the context, not really needed. Line 36: « such as cyclones an fronts », no so relevant, not really needed. Lines 57-58: « (Fitch et al., 2012) only requires », replace with : « the method used by Fitch et al. ».

We eddited the points above as suggested.

> Line 74: « the LES downscaling approach », formulation could be improved.

We changed "LES downscaling" to "meso-to-micro downscaling" as it's consitent with our previous argument and more relevant.

> Line 75: « necessary for generating turbulence », replace with something like « allows faster development of turbulence ».

We eddited the phrases as suggested.

> Lines 79-80: could be improved, e.g. with something like « an idealized case is used to compare the simulation results from WRF-SADLES with PALM ».

We revised the sentence, keeping the active voice, to "we initially conduct a comparison using an idealized case, evaluating the simulations of a 5-MW wind turbine from WRF-SADLES against those from the PALM model."

> Lines 89-90: does not look appropriate with Section 5. Section 6 is missing.

As we restructred the paper (Remove Section 5 as suggested), we revised this paragraph including the Conclusion section (now Section 5).

> Lines 97-98: « The wind speed at the rotor […] », introduce this at line 108, before actually using the wind speed at the rotor in the equation.

Thank you for your comment, for a more logical flow, we change the euation for $V = V_0(1-a)$ to $V_0 = V/(1-a)$ and move it down to before equation (5) and (6).

> L115: « we adopt », replace with « we test ».

This phrase is removed in our intensive revison of the section.

> L 144: « improve the realism of turbulent representation », not appropriate.

We changed the phrase to "accelerate the development of turbulence within the nested domain"

> L172-173: « models, known for their realistic simulation », not clear (we could think the statement also includes WRF-SADLES), improve formulation, could be simply removed.

We removed the phrase as suggested.

> L173-174: indicate also the resolution with respect to the rotor diameter », i.e. how many points per rotor diameter.

We added the information (about 12 and 4 grid points per rotor diameter)

> Line 184: « an aspect ratio of 2:1 », add the directions (could be vertical at this moment in the text).

We changed it to "a horizontal aspect ratio..."

> Line 193: « zonal direction », not clear

For clarity, we added explanation "zonal (eastward) direction"

> Line 207: reference in bad format.

We fixed the format.

> Line 222: express time in hours like for the WRF simulation.

We revised the whole section that described the PALM simulation, addressing a number of issues including this point.

> Line 230: « suggesting a dependency of momentum fluxes on model solution », not sure to follow, may need more explanation or reformulation. Line 232: « conversely », not adapted. Line 241: « methodological », maybe « numerical »? Line 245-246: improve, e.g. with something like « PALM simulation exhibits stronger wake expansion than the WRF simulation ». Line 250: « but one possibility is related », not very elegant, could be improved. Line 252: « (2D) », remove parenthesis. Line 253: « agree around 50% », not adapted, remove. Line 280: « vertical structure », modify with « «vertical profile ». Line 290: « brings them closer », not elegant.

As this whole section is revised thoroughly acording to the point above, these phrases are no longer presented in the new version.

> Line 388: « LLJ) », remove parenthesis.

We removed as suggested

> Line 398: »UTC., », remove « . ».

We editted as suggested

> Line 416: « become fully developed after about two kms », justify the statement or make less precise statement.

We make a less precise statement by changing two kilometers by a short distance.

> Line 452: « we imply », not appropriate.

We changed "imply" to "propose"

> Line 465: « model crashes », not so elegant.

We changed it to "model failures"

> In figures 4,6,7, « similar than in fig […] », copy-paste the appropriate description, each figure should be self descriptive.

We revised the figures captions and made them self-descriptive.

> Fig 5, « evaluation point » maybe not so clear (not really a point since it's a profile), « location » could be better.

We changed "point" to "location" as suggested.

> Fig 12, « radial wind speed », not so clear to me in this context.

For clarity, we swap the panel of the figure and change the caption to "(a) Line-of-sight (LOS) velocity from horizontal LiDAR scans at the FINO1 station around 23:00 UTC. (b) Simulated LOS velocity from from WRF-SADLES at 20:00 UTC."

> Fig 14: the layout in panel in (b) could be incorporated in Fig 9, and the y axis of Fig (a) and (b) could be rescaled.

Thank you for your suggestion. We incoporated the turbine layout to Fig. 9 (Now Fig. 8). Regarding the vertical scale of Fig.12. (now Fig. 11), we would like to keep as the current with the minimum on 0 to reflect the variation with their actual magnitude.

> Table 2: « real-data », not very elegant.

We changed "real-data" to "realistic"

**Remarks/questions:**

> Why only potential temperature in the perturbation method and not the wind speed components?

As explained in Muñoz-Esparza et al. 2014, the idea of cell perturbation of potential temperature is to trigger a formation of microscale three-dimensional motions, not to impose a developed turbulent field (i.e. perturbed wind speed components). Because this is not the main point of the paper, we avoid dicuss it further in the manuscipt.

Ref: Muñoz-Esparza, D., Kosoví c, B., Mirocha, J., and van Beeck, J.: Bridging the transition from mesoscale to microscale turbulence in numerical weather prediction models, Boundary-layer meteorology, 153, 409–440, 2014

> Why using a precursor simulation in PALM and no precursor in WRF simulation ?

To clarify the concern, we added a short paragraph (Lines 206-211) "To achieve a quasi-equilibrium state of turbulence, a specialized LES model such as PALM typically conducts a precursor run, often using a smaller domain to minimize computational costs. However, this precursor technique is not supported within the WRF model. Instead, we opt for a spin-up period of 20 hours solely in the outer domain."

**Responses to referee #3**

> Overall, authors addressed the comments and improved the manuscript. This article can be directly accepted after they fix the radiation issue.

Thank you for reviewing our paper. Regarding the radiation, in idealized LES with a flat surface, the radiation schemes are typically disable. Anway, we discuss your points as follows:

- In Lines 199-203. "In idealized LES, the focus is on resolving the turbulent structures within the domain. To achieve this, non-essential physical parameterization schemes, including microphysics, cumulus convection, and radiation, were disabled. The effect of surface radiation on the development of turbulence is represented by a surface turbulence heat flux of $\overline{(\theta' w')}_s = 0.02 \, \mathrm{K \, m \, s^{-1}}$, similar to some previous studies (Muñoz-Esparza et al., 2014; Kale et al., 2022). This presents a weak convective boundary layer."

- In Lines 232-232, we discussed the the point again, inclludding the two mention reference from referee #3. "Similar to WRF-SADLES, in idealized simulations of PALM, we do not use other physics parameterizations, such as microphysics, cumulus, and radiation. Although for some LES applications of urban conditions, the solar radiation processes may play an important role **(e.g., Salim et al., 2020; Krˇc et al., 2021)**, in typical LES with a flat surface like ours, the radiation parameterization is not used to focus on the main responsible processes and reduce computation (Avissar and Schmidt, 1998).. The effect of solar

radiation is instead represented by a uniform surface turbulence heat flux of $\overline{(\theta' w')}_s = 0.02$ W m s$^{-2}$, similar to WRF-SADLES's simulations."